



# OH, HO₂, and RO₂ radical chemistry in a rural forest environment: Measurements, model comparisons, and evidence of a missing radical sink

Brandon Bottorff[l], Michelle M. Lew[1a], Youngjun Woo[1], Pamela Rickly[2b], Matthew D. Rollings[3c], Benjamin Demming[3d], Daniel C. Anderson[4e], Ezra Wood[4], Hariprasad D. Alwe[5f], Dylan B. Millet[5], Andrew Weinheimer[6], Geoff Tyndall[6], John Ortega[6], Sebastien Dusanter[7], Thierry Leonardis[7], James Flynn[8], Matt Erickson[8], Sergio Alvarez[8], Jean C. Rivera-Rios[9g], Joshua D. Shutter[9h], Frank Keutsch[9], Detlev Helmig[10], Wei Wang[11], Hannah M. Allen[12], Steven Bertman[13], and Philip S. Stevens[1,2]

[1] Department of Chemistry, Indiana University, Bloomington, IN 47405, USA
[2] O'Neill School of Public and Environmental Affairs, Indiana University, Bloomington, IN 47405, USA
[3] Department of Chemistry, University of Massachusetts, Amherst, MA, 01003, USA
[4] Department of Chemistry, Drexel University, Philadelphia, PA, 19104, USA
[5] Department of Soil, Water, and Climate, University of Minnesota, Twin Cities, Saint Paul, MN, 55108, USA
[6] National Center for Atmospheric Research, Boulder, CO, 80305, USA
[7] IMT Nord Europe, Institut Mines- Télécom, Univ. Lille, Center for Energy and Enivronment, F-59000 Lille, France
[8] Department of Earth and Atmospheric Sciences, University of Houston, Houston, TX, 77004, USA
[9] Department of Chemistry and Chemical Biology, Harvard University, Cambridge, MA, 02138, USA
[10] Boulder A.I.R. LLC, Boulder, CO 80305, USA
[11] Institute of Arctic and Alpine Research, University of Colorado, Boulder, CO 80309, USA
[12] Division of Chemistry and Chemical Engineering, California Institute of Technology, Pasadena, CA, 91125, USA
[13] Institute of the Environment and Sustainability, Western Michigan University, Kalamazoo, MI, 49008, USA

[a] now at: California Air Resources Board, Sacramento, CA, USA
[b] now at: Colorado Department of Public Health and Environment, Denver, CO, USA
[c] now at: Dept. of Chemistry, University of California, Berkeley CA, USA
[d] now at: Dept. of Chemistry, Smith College, Northampton MA, USA
[e] now at: GESTAR II, University of Maryland Baltimore County, Baltimore, MD, USA
[f] now at: Forschungszentrum Jülich, Institute of Energy and Climate Research, Troposphere (IEK-8), Jülich, Germany
[g] now at: School of Chemical & Biomolecular Engineering, Georgia Institute of Technology, Atlanta, GA, USA
[h] now at: Department of Soil, Water and Climate, University of Minnesota, St. Paul, MN USA

*Correspondence to*: Brandon Bottorff (brapbott@indiana.edu); Philip S. Stevens (pstevens@indiana.edu)

**Abstract.** The hydroxyl (OH), hydroperoxy (HO₂), and organic peroxy (RO₂) radicals play important roles in atmospheric chemistry. In the presence of nitrogen oxides (NOₓ), reactions between OH and volatile organic compounds (VOCs) can initiate a radical propagation cycle that leads to the production of ozone and secondary organic aerosols. Previous measurements of these radicals under low-NOₓ conditions in forested environments characterized by emissions of biogenic VOCs, including isoprene and monoterpenes, have shown discrepancies with modeled concentrations.

During the summer of 2016, OH, HO₂ and RO₂ radical concentrations were measured as part of the Program for Research on Oxidants: Photochemistry, Emissions, and Transport – Atmospheric Measurements of Oxidants in Summer (PROPHET-AMOS) campaign in a mid-latitude deciduous broadleaf forest. Measurements of OH and HO₂ were made by laser-induced fluorescence – fluorescence assay by gas expansion techniques (LIF-FAGE) and total peroxy radical (XO₂) mixing ratios were measured by an ethane chemical amplification (ECHAMP) instrument. Supporting measurements of photolysis frequencies, VOCs, NOₓ, O₃, and meteorological data were used to constrain a zero-dimensional box model utilizing either the Regional Atmospheric Chemical Mechanism (RACM2), or the Master Chemical Mechanism (MCM). Model simulations tested the influence of HOₓ regeneration reactions within the isoprene oxidation scheme from the Leuven Isoprene Mechanism (LIM1). On average, the LIM1 models overestimated daytime maximum measurements by approximately 40% for OH, 65% for HO₂, and more than a



factor of two for $XO_2$. Modelled $XO_2$ mixing ratios were also significantly higher than measured at night. Addition of $RO_2 + RO_2$ accretion reactions for terpene-derived $RO_2$ radicals to the model can partially explain the discrepancy between measurements and modelled peroxy radical concentrations at night but cannot explain the daytime discrepancies when OH reactivity is dominated by isoprene. The models also overestimated measured concentrations of isoprene-derived hydroxyhydroperoxides (ISOPOOH) by a

factor of ten during the daytime, consistent with the model overestimation of peroxy radical concentrations. Constraining the model to the measured concentration of peroxy radicals improves the agreement with the measured ISOPOOH concentrations, suggesting that the measured radical concentrations are more consistent with the measured ISOPOOH concentrations. These results suggest that the models may be missing an important daytime radical sink and could be overestimating the rate of ozone and secondary product formation in this forest.

**1 Introduction**

As a dominant oxidant in the lower troposphere, the hydroxyl radical (OH) initiates reactions with volatile organic compounds (VOCs) leading to the production of hydroperoxy radicals ($HO_2$) and organic peroxy radicals ($RO_2$). In the presence of nitrogen oxides ($NO_x = NO + NO_2$), reactions of these radicals establish a fast cycle that can produce ozone and secondary organic aerosols (SOA). Given their central role in atmospheric chemistry, an accurate understanding of radical chemistry is important to address

current issues of air quality and climate change. Because of their short atmospheric lifetimes, measurements of these radicals can provide a test of our understanding of this complex chemistry, including our knowledge of radical sources, sinks, and propagation pathways (Heard and Pilling, 2003).

Several field campaigns have been conducted to investigate radical concentrations in both urban and forested environments. Although measurements of OH concentrations in urban areas have been generally consistent with model predictions

(Ren et al., 2003; Shirley et al., 2006; Kanaya et al., 2007a; Dusanter et al., 2009b; Lu et al., 2013; Griffith et al., 2016; Tan et al., 2017; Tan et al., 2018; Tan et al., 2019; Whalley et al., 2021) measurements of peroxy radicals in such environments have generally been underpredicted by atmospheric models (Griffith et al., 2016; Baier et al., 2017; Tan et al., 2017; Whalley et al., 2021). Measurements in forested regions characterized by low $NO_x$ mixing ratios and elevated emissions of biogenic VOCs, such as isoprene and monoterpenes, have indicated discrepancies with modelled results, with several observations of higher-than-expected

OH concentrations in isoprene-rich environments (Tan et al., 2001; Lelieveld et al., 2008; Hofzumahaus et al., 2009; Whalley et al., 2011; Lu et al., 2012; Rohrer et al., 2014). However, several recent studies have revealed potential interferences with measurements of OH radicals in forested environments (Mao et al., 2012; Novelli et al., 2014b; Feiner et al., 2016; Lew et al., 2020). Accounting for these interferences resulted in measured OH concentrations that were in good agreement with model predictions in these forested areas.

In contrast, measurements of $HO_2$ and $RO_2$ radical concentrations in forested areas have shown variable agreement with model predictions. In these environments, measured $HO_2$ concentrations were sometimes found to agree with model predictions (Tan et al., 2001; Ren et al., 2006; Feiner et al., 2016; Tan et al., 2017), but were sometimes lower (Carslaw et al., 2001; Kanaya et al., 2007b; Whalley et al., 2011; Kanaya et al., 2012; Mao et al., 2012; Griffith et al., 2013; Mallik et al., 2018), or higher than model predictions (Carslaw et al., 2001; Kubistin et al., 2010; Kim et al., 2013; Hens et al., 2014). Part of this variability may be

due to measurement interferences from certain $RO_2$ radicals in systems that detect $HO_2$ through the conversion to OH using the $HO_2 + NO \rightarrow OH + NO_2$ reaction (Fuchs et al., 2011; Whalley et al., 2013; Hens et al., 2014; Crowley et al., 2018; Lew et al., 2018). However, the extent of $RO_2$ radical contributions to $HO_2$ measurements in many of the earlier campaigns mentioned above





is not clear. While accounting for this interference would improve agreement when the model underestimates HO$_2$, it would worsen agreement in the case of an overestimation.

85       The discrepancies between measured and modeled radical concentrations in forest environments brings into question our understanding of the chemistry of biogenic VOCs (BVOCs) and their contribution to the production of ozone and SOA in the atmosphere. Isoprene is of particular importance due to its global abundance and high reactivity with the OH radical (Wennberg et al., 2018). Current models suggest that emissions of isoprene alone account for half of global non-methane VOC emissions (Guenther et al., 2012; Wennberg et al., 2018). Several theoretical and laboratory studies have investigated the atmospheric

chemistry of isoprene and its oxidation products, revealing that isomerization of isoprene-based peroxy radicals and subsequent product pathways could recycle OH and HO$_2$ radicals resulting in higher radical concentrations under low-NO$_x$ conditions (Fuchs et al., 2013; Peeters et al., 2014; Liu et al., 2017; Wennberg et al., 2018).

      In addition to isoprene, other biogenic VOCs, including monoterpenes, can play a significant role in the overall oxidative capacity of some environments. Globally, monoterpene emissions are estimated to be more than 100 Tg yr$^{-1}$ and constitute as much

as 10% of BVOC emissions (Sindelarova et al., 2014). While emissions of isoprene are strongly dependent on photosynthetic photon flux as well as temperature, several plant species emit monoterpenes also under dark conditions (Harley et al., 1996; Owen et al., 2002). Similar to the chemical mechanism of isoprene oxidation, peroxy radicals produced from the oxidation of monoterpenes can undergo isomerization reactions as part of autooxidation mechanisms, leading to the production of highly-oxidized peroxy radical products (Jokinen et al., 2014). Under low-NO$_x$ conditions, these reactions can compete with reaction with

NO as well as with peroxy radical self and cross reactions.

      While it is known that self- and cross-reactions of RO$_2$ can form either alkoxy radicals (reaction R1) or an alcohol and a carbonyl species (R2) (Orlando and Tyndall, 2012), a third pathway that leads to the formation of a dimeric dialkyl peroxide (R3) has been proposed but previously regarded as less significant due to low yields for small RO$_2$ species (Lightfoot et al., 1992; Tyndall et al., 2001; Noell et al., 2010). However, recent studies have observed the formation of gas phase C$_{19}$-C$_{20}$ dimer

compounds and suggest that autoxidation and RO$_2$ + RO$_2$ reactions between terpene-derived peroxy radicals may form low-volatility dimers or accretion products (R3) (Crounse et al., 2013; Ehn et al., 2014; Berndt et al., 2018a; Berndt et al., 2018b; Bianchi et al., 2019).

$$RO_2 + R'O_2 \longrightarrow RO + R'O + O_2 , \tag{R1}$$

$$RO_2 + R'O_2 \longrightarrow ROH + R'C = O + O_2 , \tag{R2}$$

$$RO_2 + R'O_2 \longrightarrow ROOR' + O_2 , \tag{R3}$$

In addition to significantly affecting of SOA formation, these reactions could be relevant alongside reactions with NOx or HO2 as radical termination reactions and should be considered when modelling radical concentrations in low-NOx regions characterized by significant biogenic VOC emissions.

      This study presents measurements of OH, HO2, and total peroxy radical (XO2 = RO2 + HO2) concentrations made within

a remote forested region during the PROPHET-AMOS 2016 (Program for Research on Oxidants: Photochemistry, Emissions, and Transport – Atmospheric Measurements of Oxidants in Summer) field campaign. The measurements are compared to predicted radical concentrations from zero-dimensional box models constrained to a wide range of trace gases and meteorological conditions. Additional model simulations that incorporate the Leuven Isoprene Mechanism (LIM1) for isoprene degradation and a series of RO2 + R'O2 reactions that form accretion products are accompanied by a radical budget analysis to test current atmospheric

chemistry mechanisms and investigate the fate of isoprene- and monoterpene-derived peroxy radicals in this low-NOx environment.



## 2 Experimental Methods

### 2.1 PROPHET-AMOS measurement site

All measurements described below were performed as part of the PROPHET-AMOS 2016 field campaign. Measurements were
conducted throughout the month of July at the PROPHET facility at the University of Michigan Biological Station (UMBS) in
northern Michigan (45.5588° N, 84.7145° W). The mixed deciduous and coniferous forest site consists primarily of isoprene-
emitting species such as big-tooth aspen and red oak but also monoterpene-emitting species such as red maple, white pine, and
paper birch (Ortega et al., 2007; Bryan et al., 2015). The site has been described in more detail elsewhere (Carroll et al., 2001;
Ortega et al., 2007; Griffith et al., 2013). The majority of the measurements described below were performed near the top of the
31-m tower, approximately 10 m above the forest canopy either by placing the instrument directly on the top of the tower, sampling
from a glass manifold in the laboratory that pulled air from the top of the tower, or by sampling from individual inlets from the top
of the tower. Measurements of ozone were taken from the top of the nearby Ameriflux tower, which is 100 m to the north of the
PROPHET tower. Table 1 summarizes the measurements used in this study.

**Table 1: Measured species used for data analysis and model calculations and respective measurement techniques.**

| Measured Species | Instrument | Technique | Reference | LOD |
|---|---|---|---|---|
| OH, HO$_2$ | LIF-FAGE | Laser-induced fluorescence-fluorescence assay by gas expansion | (Dusanter et al., 2009a; Griffith et al., 2013) | OH - $6.5 \times 10^5$ cm$^{-3}$ (2 hour) HO$_2$ - $1.1 \times 10^7$ cm$^{-3}$ (0.4 ppt) (20 s) |
| XO$_2$ | ECHAMP | Ethane Chemical amplification | (Wood and Charest, 2014; Wood et al., 2017) | 1-3 ppt (2 min) |
| NO, NO$_2$ | 2-channel chemiluminescence | Chemiluminescence and LED converter for NO$_2$ | NCAR (Ridley and Grahek, 1990) | |
| O$_3$ | Thermo Scientific 49C | UV absorbance | | 1.0 ppb |
| VOCs | PTR-Qi-ToF | Proton Transfer Reaction-Time-of-Flight Mass Spectrometry | (Millet et al., 2018) | |
| NMHCs | Online GC/FID/FID | Gas chromatography with flame ionization detection | (Badol et al., 2004) | 10-100 ppb (1.5 h) |
| OVOCs | DNPH-HPLC | dinitrophenylhydrazine cartridges and offline high performance liquid chromatography and UV detection | | |
| jNO2 | | spectral radiometry | (Shetter and Müller, 1999) | $0.3 \times 10^{-4}$ s$^{-1}$ |
| ISOPOOH | GC-HRToF-CIMS | Low-pressure gas chromatography coupled with high-resolution time-of-flight chemical ionization mass spectrometry | (Vasquez et al., 2018) | ~10 ppt |

During the campaign, isoprene, the sum of methylvinylketone and methacrolein, monoterpenes, acetaldehyde, and other
VOCs and oxygenated VOCs (OVOCs) were measured by the University of Minnesota using proton transfer reaction-quadrupole
interface time-of-flight mass spectrometry (PTR-QiTOF) (Millet et al., 2018). In addition, C2–C10 alkanes and alkenes, butadiene,
C6–C9 aromatic compounds, and isoprene were measured by IMT Nord Europe using a thermal desorption gas chromatography



with flame ionization detection (GC-FID) instrument with a 1.5-h time resolution, while C2–C10 aldehydes, C2–C6 ketones, and C2–C4 alcohols were measured by thermal desorption GC-FID with mass spectrometry (GC-FID-MS) with a 1.5-h time resolution

(Badol et al., 2004; Roukos et al., 2009). NO and $NO_2$ were measured by the NCAR single-channel chemiluminescence instrument (Ridley and Grahek, 1990), ozone was measured by UV absorption by the University of Colorado, and CO by laser-based off-axis integrated cavity output spectroscopy (Los Gatos Research) by the University of Houston and Rice University groups. Isoprene hydroxy hydroperoxides (ISOPOOH) were measured by a gas chromatograph chemical ionization mass spectrometer (GC-ToF-CIMS) (Vasquez et al., 2018). Photolysis frequencies were measured using spectral radiometry (Shetter and Müller, 1999) by the

University of Houston. Measurements of OH, $HO_2$ and $XO_2$ radicals are described in detail below.

## 2.2 Measurements of $HO_x$ concentrations

Both OH and $HO_2$ were measured using the Indiana University laser-induced fluorescence-fluorescence assay by gas expansion (IU-FAGE) instrument that has been described in more detail previously (Dusanter et al., 2009a; Griffith et al., 2013; Lew et al., 2020). Briefly, OH radicals are detected by laser-excitation at 308 nm and subsequent fluorescence detection also at 308 nm. The

sampled air expands into a low-pressure cell, which extends the OH fluorescence lifetime by reducing the concentration of species that may quench OH fluorescence and allows temporal filtering of OH fluorescence from more intense scattered laser light (Heard and Pilling, 2003).

The IU-FAGE laser system used in this study consisted of a Spectra Physics Navigator II YHP40-532Q that produced approximately 7.5 W of 532-nm radiation (10-kHz repetition rate) to pump a Sirah Credo dye laser (255 mg/L of Rhodamine 610 and 80 mg/L of Rhodamine 101 in ethanol) resulting in approximately 40 mW of radiation that is tunable near 308 nm. This laser

system was housed in the laboratory at the bottom of the PROPHET measurement tower and 308-nm radiation was focused onto the entrance of a 50-m optical fiber to transmit the laser emission to the sampling cell.

The IU-FAGE sampling cell was located atop the 31-m measurement tower, approximately 10 meters above the forest canopy. Ambient air was drawn into the detection cell through a pinhole inlet (0.64 mm diameter) by means of three scroll pumps

(Edwards XDS 35i) connected in parallel. The pumps were located at the bottom of the tower and connected to the sampling cell by two parallel 3.8-cm inner-diameter vacuum hoses, which resulted in a sampling cell pressure of 0.6 kPa (4.5 torr) and a flow of 3 SLPM through the sampling inlet.

On average, approximately 1.25 mW of 308-nm radiation exited the 50-m fiber and entered the sampling cell during the campaign. The laser emission enters the sampling cell perpendicular to the sampled air mass and intersects the expanded air in a

White cell configuration with approximately 24 passes. The OH fluorescence is collected along an axis that is orthogonal to both the laser emission and sampled air mass and detected using a microchannel plate photomultiplier tube (MCP-PMT) detector (Hamamatsu R5946U), a preamplifier (Stanford Research Systems SR445), and a photon counter (Stanford Research Systems SR400). The MCP-PMT is turned off and the photon counter is inactive during the laser pulse by means of a delay generator (Berkley Nucleonics 565) to allow the OH fluorescence to be temporally filtered from scattered laser light.

The net OH fluorescence signal is determined through successive spectral-modulation cycles in which the dye laser emission wavelength is tuned on- and off- resonance with the $Q_1(3)$ transition of OH near 308 nm. A background signal, which primarily consists of scattered laser light that extends into the detection window, is established by tuning the laser emission off-resonance with the OH transition, and therefore not exciting OH radicals. This background signal is subtracted from on-resonance signal. A reference cell in which OH is generated by the thermal dissociation of water vapor is used to ensure maximum overlap

between dye-laser emission and the OH transition wavelength.





The IU-FAGE measurements of OH are subject to potential interferences when OH radicals are generated inside the detection cell. In the presence of water vapor, the photolysis of ozone by the laser can produce hydroxyl radicals through reactions R4 and R5 (Davis et al., 1981a; Davis et al., 1981b).

$$O_3 + hv \ (< 340 \text{ nm}) \rightarrow O(^1D) + O_2 \ , \tag{R4}$$

$\quad O(^1D) + H_2O \rightarrow OH + OH, \tag{R5}$

To characterize this and any other interference, a chemical scrubbing technique is used to remove ambient OH prior to entering the detection cell (Griffith et al., 2016; Rickly and Stevens, 2018; Lew et al., 2020). This chemical modulation technique is used to monitor levels of the laser-generated ozone-water interference and any other interference that may produce OH radicals inside the detection cell. Hexafluoropropylene ($C_3F_6$, 95.5% in $N_2$; Matheson Gas) was added through a circular injector 1 cm above the

inlet with a flow rate of approximately 3.5 sccm to remove 95% of external OH radicals (Rickly and Stevens, 2018). The differences between the measured OH during $C_3F_6$ addition and OH measurements including the interference represent the net ambient OH concentration in the atmosphere. The addition of $C_3F_6$ is modulated in between ambient OH measurements every 15 min for a duration of 10 min.

Measurements of $HO_2$ were made indirectly after addition of NO to the sampled air mass to convert ambient $HO_2$ to OH

through the fast $HO_2 + NO \rightarrow OH + NO_2$ reaction. A small flow (approximately 2 sccm) of NO (Matheson, 1% in nitrogen) was added to the sampled air mass through a Teflon loop injector that was positioned directly below the sampling inlet, resulting in an added NO concentration of approximately $9 \times 10^{11}$ cm$^{-3}$. The fraction of $HO_2$ converted into OH was measured during calibration experiments performed during and after the campaign and was $14.0 \pm 3.2\%$. This low NO concentration minimized the impact of interferences from $RO_2$ radicals derived from the OH-initiated oxidation of alkenes and aromatics that can be quickly converted to

$HO_2$ (Fuchs et al., 2011; Lew et al., 2018). The high conversion efficiencies reported for these $RO_2$ radicals is due to the rapid decomposition of β-hydroxyalkoxy radicals that are formed from the $RO_2 + NO$ reaction. This decomposition forms a hydroxyalkyl radical that reacts rapidly with $O_2$ to produce $HO_2$ in the detection cell. This can lead to the detection of both $HO_2$ and a fraction of $RO_2$ radicals denoted as $HO_2^*$ ($HO_2^* = HO_2 + \alpha RO_2$, $0<\alpha<1$). Calibrations before and after the campaign similar to those described in Lew et al. (2018) indicated that the low NO concentration injected into the detection cell (approximately $9\times10^{11}$ cm$^{-}$

$^3$) resulted in an $RO_2$-to-$HO_2$ conversion efficiency of approximately 10% for isoprene-based peroxy radicals and an $RO_2$-to-OH conversion efficiency of less than 2% (Fig. S1). As a result, the $HO_2$ measurements were performed at the low NO flow that effectively minimized the impact of any potential interference from isoprene-derived $RO_2$ species that are dominant during the day at the PROPHET site (Griffith et al., 2013).

The instrument was calibrated by producing known concentrations of OH and $HO_2$ from the photolysis of water vapor in

air as described by Dusanter et al. (2008). The calibration source consists of an aluminum flow reactor with quartz windows on two opposite sides. Aluminum cartridges adjacent to each window house a low-pressure mercury pen lamp and a photodiode detector, both of which are continuously purged with dry nitrogen to stabilize the lamp-temperature and prevent light absorption by atmospheric gases. Radiation from the mercury lamp passes through a bandpass filter centered at 185 nm prior to illuminating the flow reactor and detector. The location of the mercury lamp and photodiode is adjustable along the length of the calibration

source to allow for the measurement of radical surface loss between the illuminated region and the exit of the calibrator. For calibrations during PROPHET, zero air was delivered to the calibration source at a flow rate of 50 L min$^{-1}$. A variable fraction of the flow (5 – 40%) was diverted through a set of custom bubblers containing high purity water at the base of the tower. This humidified fraction of air was mixed back with the initial flow in approximately 35 m of PTFE tubing (1.25 cm i.d.) before entering



the calibration source. Calibrations were performed before, after, and intermittently during the campaign to track changes in sensitivity. The uncertainty associated with this calibration technique is approximately 18% (1σ) for both OH and HO₂.

As previously mentioned, a 50-m fiber optic cable was used to transmit laser radiation to the sampling cell for the above-canopy measurements. The long fiber presented technical challenges that impacted the performance of the IU-FAGE instrument. Due to the length of the fiber, the laser pulse was temporally broadened prior to entering the detection cell and resulted in an increase of background laser scatter of the instrument. This broadened pulse made temporal filtering of scattered laser light

difficult, and ultimately led to a lower sensitivity and a higher limit of detection for OH. In addition, the length of the fiber corresponded to a decrease in transmission of radiation through the fiber. An average transmission of 8% led to 0.76-2.15 mW of 308 nm radiation in the detection cell over the course of the campaign. Due to low laser power and high background signal, long averaging times were necessary for OH measurements. The limit of detection for OH was $6.5 \times 10^5$ molecules cm⁻³ (1σ, 2-hour average). Measurements of HO₂ were performed approximately once per hour with a limit of detection of $1.1 \times 10^7$ cm⁻³ (0.4 ppt)

(1σ, 20-s average).

**2.3 Measurements of Total Peroxy Radicals (XO₂)**

Total peroxy radicals were measured by an Ethane Chemical Amplifier (ECHAMP) instrument that has been previously described in detail (Wood et al., 2017). This instrument is similar to traditional chemical amplifiers that mix ambient air with excess CO and NO (Cantrell and Stedman, 1982; Hastie et al., 1991; Cantrell et al., 1996), but instead utilizes chemical amplification by ethane

(C₂H₆) and NO followed by detection of NO₂ using cavity-attenuated phase-shift (CAPS) spectroscopy.

The ECHAMP inlet box was positioned on the top platform of the tower at a height of 31 m. Ambient air was sampled at a flow rate of 7.3 SLPM through a 0.4 cm inner diameter (ID) glass inlet that was internally coated with halocarbon wax to minimize radical loss on surfaces. A small flow (0.35 SLPM) of pure O₂ was added through a side port to this main flow. The O₂ addition increases the O₂ mixing ratio in the sampled air to 24.6% and reduces both the value and the variability of the relative

humidity in the sampled air. The sampled air finally entered two reaction chambers at individual flow rates of 1.0 L min⁻¹ with the remaining sampled air used to monitor temperature and RH. In the amplification chamber, the sampled air was immediately mixed with 20 sccm of 50 ppm NO and 20 sccm of 50% C₂H₆ through an upstream reagent addition port, leading to final mixing ratios for NO and C₂H₆ of 1 ppm and 1% respectively. A flow of 20 sccm N₂ was added downstream, 100 ms later. In this chamber, ROₓ species are converted to HO₂ and OH through reactions with NO (R6 – R8). Reactions (R9 – R12) repeat several times leading to

the formation of NO₂ that is subsequently measured by a CAPS monitor.

$$RO_2 + NO \longrightarrow RO + NO_2 , \tag{R6}$$

$$RO + O_2 \longrightarrow HO_2 + products, \tag{R7}$$

$$HO_2 + NO \longrightarrow OH + NO_2, \tag{R8}$$

$$OH + C_2H_6 \longrightarrow C_2H_5 + H_2O, \tag{R9}$$

$$C_2H_5 + O_2 + M \longrightarrow C_2H_5O_2 + M, \tag{R10}$$

$$C_2H_5O_2 + NO \longrightarrow C_2H_5O + NO_2 , \tag{R11}$$

$$C_2H_5O + O_2 \longrightarrow CH_3CHO + HO_2, \tag{R12}$$

In the background chamber, the sampled air was first mixed with NO and N₂, and then C₂H₆ was added 100 ms later. In this mode, ambient radicals are removed by successive reactions with NO (R6 – R8) until they form HONO via the OH + NO →



HONO reaction, and therefore amplification chemistry does not occur. After reagent addition, air from each chamber enters identical CAPS monitors. The CAPS $NO_2$ measurements from the background chamber represent ambient $NO_2$, $NO_2$ from the reaction of ambient $O_3$ with added NO, and $NO_2$ from reactions of ambient peroxy radicals with NO, but not from ethane amplification reactions. The CAPS $NO_2$ monitor following the amplification chamber measures the sum of that observed from the background chamber and $NO_2$ produced from amplification chemistry. The amount of $NO_2$ produced from amplification reactions

($\Delta NO_2$) is determined from the difference between the amplification and background chambers. The concentration of peroxy radicals is calculated by dividing [$\Delta NO_2$] by the experimentally determined amplification factor, $F$.

$$[RO_x] = \Delta[NO_2]_{(CAPS_{RO_x} - CAPS_{O_x})}/F, \tag{1}$$

The amplification factor was determined as a function of relative humidity by producing known concentrations of peroxy radicals with two different calibration sources. The first source relies on the photolysis of water vapor method which is similar to

that described above for the IU-FAGE instrument and is commonly used to calibrate other chemical amplifiers (Mihele and Hastie, 2000; Horstjann et al., 2014) and LIF-FAGE instruments (Heard and Pilling, 2003; Dusanter et al., 2008). This method produces equivalent concentrations of OH and $HO_2$ that are quantified by $O_2$ actinometry and measured concentrations of $H_2O$ and $O_3$ in the calibration gas, and OH can be quantitatively converted to $HO_2$ or isoprene peroxy radicals through the addition of $H_2$ or isoprene, respectively. The second calibration source was based on the photolysis of methyl iodide ($CH_3I$) at 254-nm to produce $CH_3O_2$

radicals (Anderson et al., 2019). The radical concentration is quantified by reaction with NO, in the absence of ethane, to produce $NO_2$ that is measured by CAPS. During PROPHET the ECHAMP limit of detection was 1-3 ppt ($2\sigma$, 2-min average). During PROPHET, the $CH_3I$ calibration method was used as the primary source and the water vapor photolysis method was used less frequently to quantify the relative response of ECHAMP to $HO_2$ and $CH_3O_2$ radicals (see below).

As described in Wood et al. (2017) and Kundu et al. (2019), ECHAMP does not detect all peroxy radicals with equal

sensitivity. A portion of $RO_2$ radicals are converted to alkyl nitrites (RONO) and alkyl nitrates ($RONO_2$) via association reactions with NO and a portion of all sampled radicals are lost to wall reactions. Wall loss rate constants measured in the laboratory for halocarbon-coated 0.4 cm ID glass were typically 1.6 $s^{-1}$ for $HO_2$ at 60% RH and less than 0.2 $s^{-1}$ for $CH_3O_2$, with isoprene peroxy radical wall loss rate constants between those two values (Kundu et al., 2019). For the sampling conditions during PROPHET (7.3 SLPM flow rate, 13 cm inlet length) this suggests only 2% of $HO_2$ was lost to wall reactions. Furthermore, an expected 8% of

isoprene peroxy radicals are lost to formation of organic nitrates. The relative sensitivity of ECHAMP to $HO_2$ radicals and $CH_3O_2$ radicals was quantified after the campaign by comparing its response to both types of radicals prepared at equal concentrations using the water vapor photolysis method. These measurements showed that the response to $HO_2$ was 2% lower than the instrument response to $CH_3O_2$. In the absence of sampling losses we would expect that the response to $CH_3O_2$ would be 10% lower than the response to $HO_2$ due to formation of $CH_3ONO$ (Wood et al., 2017). These results indicate that sampling losses of $HO_2$ were more

likely 10% and almost equal to the loss of $CH_3O_2$ due to $CH_3ONO$ formation. Further details of a calibration source comparison between the LIF and ECHAMP instruments are provided in the Supplement.

**2.4 Modeling concentrations of OH, $HO_2$, and $XO_2$**

Ambient concentrations of OH, $HO_2$, and $XO_2$ were modelled with the Master Chemical Mechanism (MCM) (Jenkin et al., 1997; Jenkin et al., 2015) and the Regional Atmospheric Chemistry Mechanism version 2 (RACM2) (Goliff et al., 2013). The

RACM2 mechanism groups several species according to their reactivity and includes more than 350 reactions. While the near-explicit MCM is expected to better represent the complex oxidation chemistry of this environment, the grouped RACM model is more computationally efficient and simpler to use in a radical budget analysis. Due to the limited isoprene oxidation mechanism



in the base RACM2 model, a series of reactions described by Tan et al. (2017) was incorporated based on the LIM1 mechanism proposed by Peeters et al. (2009; 2014). The resulting condensed version of LIM1 includes updated bulk reaction rate constants
for the 1,6-H shift isomerization reactions of the isoprene peroxy radicals as parameterized by Peeters et al. (2014). These isomerization reactions lead to the formation of $HO_2$ and hydroperoxyaldehydes (HPALDS) which can photolyze leading to OH production, as well as dihydroperoxy-carbonyl peroxy radicals (di-HPCARPs) which can rapidly decompose to produce additional OH radicals (Teng et al., 2017; Wennberg et al., 2018).

The Master Chemical Mechanism provides a near-explicit mechanism that describes the gas-phase chemical processes
involved in the degradation of over 140 VOCs. Model simulations utilized both MCM version 3.2 and MCM version 3.3.1, the latter of which incorporates the explicit LIM1 mechanism and includes the equilibrium between different isoprene peroxy radical isomers and the H-shift isomerization reactions of specific isomers resulting in $HO_x$ radical recycling through the production of HPALDs as well as di-HPCARP radicals (Jenkin et al., 2015). In this mechanism, the equilibrium rate coefficients between different peroxy radical isomers were increased and the 1,6 H-shift isomerization rate constants were decreased in order to match
early experimental results of Crounse et al. (2014) (Peeters, 2015)These changes resulted in effective bulk 1,6 H-shift peroxy radical isomerization rate constants in MCM 3.3.1 that are approximately a factor of 5 lower than the original LIM1 recommended rates (Novelli et al., 2020).

Each of the chemical mechanisms were embedded into the Framework for 0-D Atmospheric Modeling (F0AM) (Wolfe et al., 2016) to calculate radical concentrations predicted by each mechanism. Modeled chemistry for both mechanisms was
constrained to measurements of meteorological data and a wide variety of trace gas mixing ratios that were measured during the campaign (Tables S1 and S2). Model simulations were performed with a 30-min integration time and a 5-day spin-up to allow sufficient time to generate unmeasured secondary oxidation products. A 24-h lifetime was assumed for all calculated species to simulate loss via dry deposition and to prevent unexpected accumulation of some unmeasured species. Similar to Ren et al. (2013) and Lu et al. (2012), model sensitivity runs indicate that increasing this depositional loss by a factor of 2 results in changes of the
modeled $HO_2$ concentration of less than 4%. Measurement constraints were synchronized to 30-min time intervals. Species that were measured more frequently were averaged to 30-min intervals, and linear interpolation was used for species measured with lower time resolution.

In cases when speciated measurements or complete measurement sets were not available for species important to radical chemistry, an appropriate correlation analysis, or an average of previous measurements conducted at the PROPHET location were
used to constrain the model. For example, measurements of the sum of methyl vinyl ketone and methacrolein (MVK + MACR) were available throughout the campaign from the University of Minnesota's PTR-QiToF instrument, but speciated MACR was measured on some days by the IMT Nord Europe online GC/FID/FID. An MVK:MACR ratio of 0.65:0.35 was derived from a correlation of the available measurements and used to constrain the model when speciated measurements were not available. Similarly, as the sum of monoterpenes was measured by PTR-QiTOF, the mixing ratio of total monoterpenes was constrained as
α-pinene in model simulations. Because measurements of HONO concentrations at the top of the tower were unavailable, the model was constrained to the campaign average of previous measurements at this site (Griffith et al., 2013). Photolysis frequencies were calculated using a trigonometric parameterization based on solar zenith angle (Jenkin et al., 1997; Wolfe et al., 2016) and scaled according to measured values of $J(NO_2)$ to account for cloud coverage. The model uncertainty is estimated to be 30% based on uncertainties from model constraint inputs and the measured rate constants for each reaction (Griffith et al., 2013; Wolfe et al.,
325  2016).

In addition to the standard RACM2 and MCM 3.2 models, and the expanded isoprene chemistry in RACM2-LIM1 and MCM 3.3.1, a third set of model simulations were conducted to investigate the influence of $RO_2 + RO_2$ accretion reactions and dimer



formation on overall radical concentrations. A set of reactions were added to both RACM2-LIM1 and MCM 3.3.1 to create overall mechanisms (RACM-ACC and MCM-ACC) that incorporate $RO_2 + RO_2$ accretion reactions for both isoprene- and monoterpene-

based peroxy radicals. Several studies have reported observations of highly oxidized $C_{19-20}H_{28-32}O_{10-18}$ dimer products in chambers (Ehn et al., 2014) and in field measurements (Yan et al., 2016; Zha et al., 2018), suggesting that $RO_2$ reactivity in the process of dimer formation increases along with functionalization and size of the $RO_2$ radical (Berndt et al., 2018a; Berndt et al., 2018b). Rate constants for the added reactions were based on measurements from Berndt et al. (2018a; 2018b) and are intended to represent complex autooxidation and dimer formation chemistry into a model process that results in net radical termination.

Rate constants for several $RO_2 + RO_2$ reactions used in this study are shown in Table 2. As described above, measurements of the sum of all monoterpenes were interpreted as α-pinene in the model and thus rate constants measured in an exclusively α-pinene system (Berndt et al., 2018a) were used and provide only an estimation of the terpene chemistry at the PROPHET site that also includes emissions of β-pinene, limonene, and others (Carroll et al., 2001; Ortega et al., 2007). In addition to $C_{10}$-$RO_2$ radicals derived from monoterpenes, measured rate constants for self- and cross-reactions of $C_5$-$RO_2$ radicals derived from isoprene are

also included, as well as more general, slower reactions between $C_{10}$-$RO_2$ and other smaller $RO_2$ species. For the purposes of this study, rate constants are based on measurements of the least oxidized $C_{10}$-$RO_2$ species described in Berndt et al. (2018a), and thus may represent a lower limit in terms of autooxidation and dimer reactions as a radical sink. As such, the goal of this model was not to provide a detailed mechanism or exact representation of chemistry, but instead to investigate the plausibility of autooxidation and dimer formation and the relative importance that the process may have as a radical termination process in low $NO_x$, high

BVOC environment.

Table 2: Summary of $RO_2 + R'O_2 \rightarrow ROOR'$ rate constants added to RACM-ACC and MCM-ACC based on Berndt et al. (Berndt et al., 2018a; Berndt et al., 2018b)

| $RO_2$ | $R'O_2$ | k ($cm^3 molecule^{-3}s^{-1}$) |
|---|---|---|
| $O_3$-monoterpene | $O_3$-monoterpene | $9.7 \times 10^{-12}$ |
| OH-monoterpene | OH-monoterpene | $3.7 \times 10^{-11}$ |
| OH-isoprene | OH-isoprene | $6.0 \times 10^{-13}$ |
| OH-isoprene | OH-monoterpene | $1.3 \times 10^{-11}$ |
| $O_3$-monoterpene | other | $1.0 \times 10^{-12}$ |
| OH-monoterpene | other | $2.0 \times 10^{-12}$ |
| other | other | $<4.0 \times 10^{-13}$ |

## 3 Results

### 3.1 Meteorological and chemical conditions

A complete suite of supporting measurements, including meteorological conditions and important chemical species that were used

as model constraints is shown in Fig. 1, and campaign average measured values of important model constraints are shown in Fig. 2. In general, weather during the campaign was sunny with intermittent clouds, with some exceptions of more overcast days (July 8, 15, 17, 24). Mixing ratios of NO, $O_3$, and photolysis rate constants were similar to those observed during previous field campaigns at the same site (Griffith et al., 2013). The maximum observed NO mixing ratio was 480 ppt on July 11, and the average peak mixing ratio of NO was approximately 115 ppt at 9:00 local time. NO mixing ratios at night were typically less than 0.5 ppt.

Average ozone mixing ratios were between 25 and 35 ppb. Maximum average daytime temperatures of 24 °C were similar to measurements at this site in 2008, but warmer than measurements at this site in 2009, resulting in average mixing ratios of isoprene





that peaked near 3 ppb at approximately 18:00, similar to that measured in 2008, but greater than that measured in 2009 (Griffith et al., 2013). Mixing ratios of anthropogenic VOCs were low with average mixing ratios of toluene and benzene below 65 and 40 ppt, respectively.

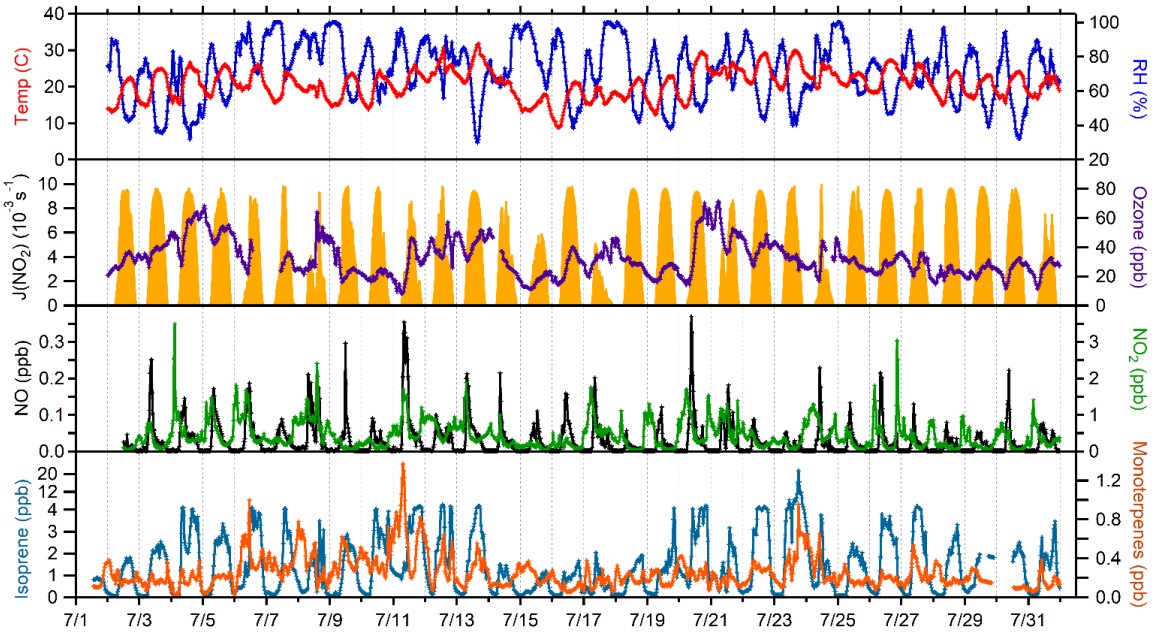


**Figure 1: Time series of measured meteorological and chemical conditions used as constraints for model calculations.**





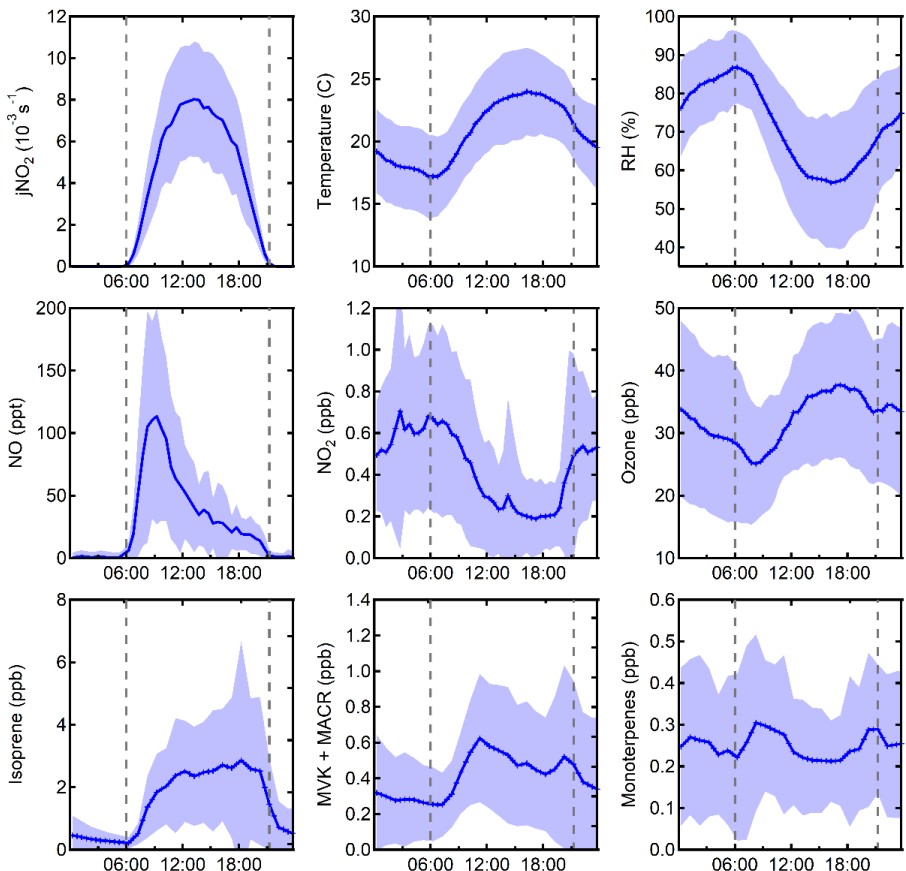

**Figure 2: Campaign average measurements of jNO₂, temperature, relative humidity, NO, NO₂, O₃, isoprene, and methyl vinyl ketone and methacrolein. Shaded areas represent the 1σ variability.**

**3.2 OH measurements and model predictions**

Measured and modelled OH, HO₂, and XO₂ concentrations from July 2 through July 31 are shown in Fig. 3 with correlation plots shown in Fig. S2. Measurements of OH were hampered by high background signals and limited laser power. Diurnal profiles with a 2-h time resolution of the OH measurements are shown in Fig. 4, in addition to the model results. An average of all OH measurements performed during the campaign shows a peak of $1.25 \times 10^6$ molecules cm⁻³ at 13:00. Measurements of OH during
the morning hours were significantly lower than all model calculations. The reason for this discrepancy is not clear but may be the result of participant activity on the top of the tower near the detection cell in the mornings during the campaign which may have influenced the OH measurements, although a systematic measurement error during this time cannot be ruled out.





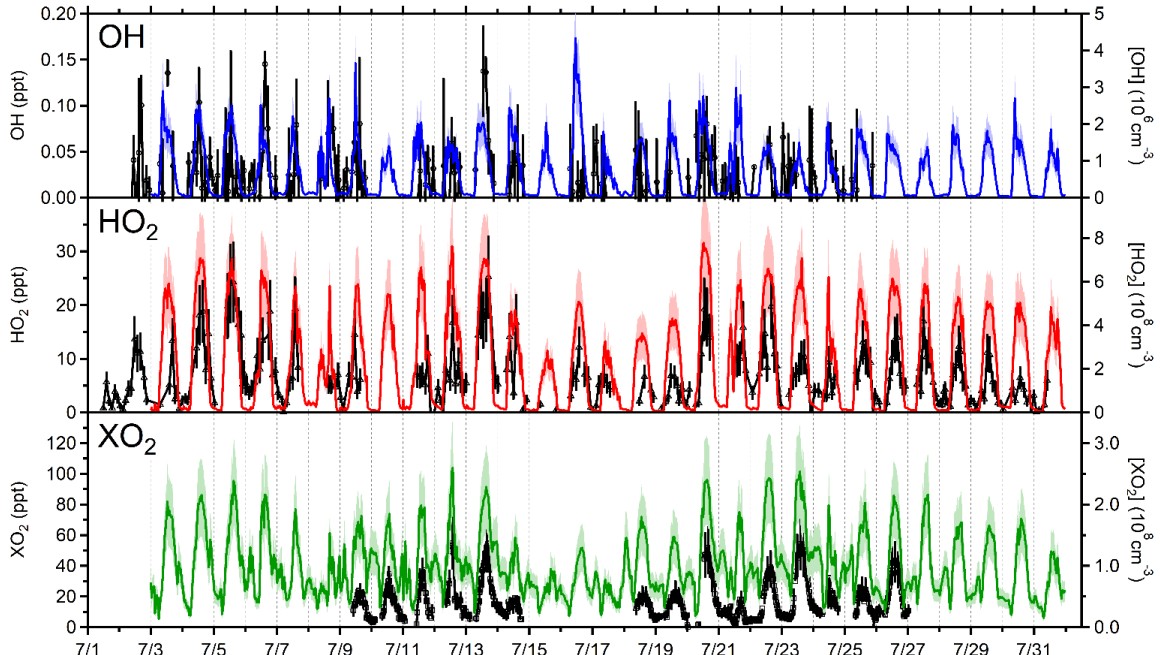

**Figure 3: Time series of radical measurements (black) and MCM v3.3.1 model predictions of OH (blue), HO₂ (red), and XO₂ (green)**
**from July 2 to July 31. Measurement of any potential interferences in the OH measurements have been subtracted and only positive OH measurements are shown for simplicity.**

Measurements of potential interferences by chemical modulation through addition of $C_3F_6$ as described above did not reveal any significant unknown interferences, similar to that observed previously at this site (Griffith et al., 2013), but in contrast to measurements by the IU FAGE instrument during the IRRONIC (Indiana Radical, Reactivity and Ozone Production
Intercomparison) campaign (Lew et al., 2020). Lew et al. (2020) found that the measured interference increased with both ozone and temperature, similar to that observed by Mao et al. (2012) who also measured a similar interference that increased with both temperature and total OH reactivity. Laboratory studies suggest that the interference could be due to the decomposition of Criegee intermediates inside the low-pressure detection cell leading to OH production (Novelli et al., 2014a; Fuchs et al., 2016; Novelli et al., 2017; Rickly and Stevens, 2018) although estimated concentrations of Criegee intermediates in similar environments are too
low to explain the observed interference (Novelli et al., 2017). Another proposed source of the interference is the decomposition of ROOOH molecules inside the FAGE detection cell formed from the reaction of OH with $RO_2$ radicals (Fittschen et al., 2019). While the sources of these interferences are still unknown, one possible explanation for the absence of a measurable interference during PROPHET-AMOS is the lower measured mixing ratios of ozone and lower temperatures compared to that measured during IRRONIC, resulting in lower mixing ratios of isoprene and other BVOCs. Based on the observed correlation of the interference
with ozone and temperature highlighted in Lew et al. (2020), it is likely that a similar interference was undetectable during PROPHET-AMOS.





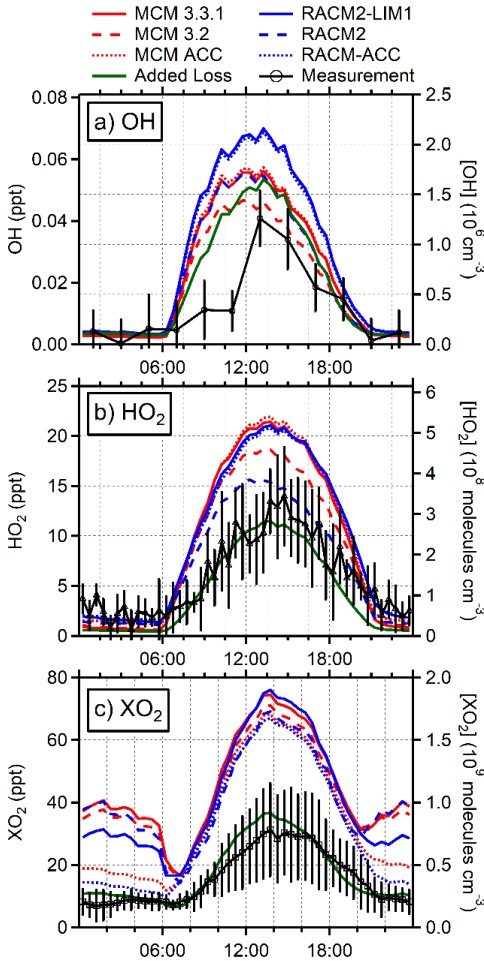

**Figure 4: Diurnal average measured (black) and modeled concentrations of (a) OH, (b) HO₂, and (c) XO₂. MCM models are shown in red and RACM in blue. The green line represents an additional version of the RACM-ACC model with added sinks for HO₂ and isoprene peroxy radicals. The colored lines represent an average of individual daily model runs from only the days that each respective species was measured (OH: 7/3–7/9, 7/11–7/14, and 7/16–7/25; HO₂: 7/3–7/9 and 7/11–7/31; XO₂: 7/9–714 and 7/18–7/26). Error bars represent the 1σ measurement precision.**

The measured OH concentrations reported here are similar to previous measurements made by the IU-FAGE instrument at the PROPHET site in 2009 but are lower than those measured at the site in 2008, although the latter measurements suffered from poor precision (Griffith et al., 2013). The results reported here are also in contrast to measurements of OH at this site in 1998 as reported by Tan et al. (2001), who reported maximum daytime concentrations of approximately $4 \times 10^6$ cm⁻³ that were approximately a factor of 3 greater than model predictions. While the mixing ratios of NOₓ and isoprene in 1998 were similar to those observed during PROPHET-AMOS, mixing ratios of ozone were higher in 1998, with the average maximum of approximately 45 ppb similar to that observed during the IRRONIC campaign (Lew et al., 2020). In addition, anomalously elevated concentrations of OH were observed at night in 1998 (Faloona et al., 2001; Tan et al., 2001). These results suggest that the OH measurements in 1998 at the PROPHET site may have been influenced by interferences similar to those observed by Mao et al. (2012), Feiner et al. (2016), and Lew et al. (2020).

As illustrated in Fig. 4, the base RACM2 and MCM 3.2 models are able to reproduce the maximum observed OH concentrations to within the combined measurement precision and uncertainty of the models. The addition of LIM1 chemistry to



the models increased the predicted average maximum OH concentration by approximately 20% between MCM 3.2 and MCM 3.3.1, and by 30% between RACM2 and RACM2-LIM1, with the MCM 3.3.1 maximum modeled OH concentrations approximately 30% greater than the measured concentrations, and the RACM2-LIM1 maximum modeled OH concentrations approximately 60% greater than the measured concentrations. These results are in contrast to several previous LIF measurements in forested environments (Rohrer et al., 2014), in which measured OH concentrations were significantly higher than modeled predictions. However, the results reported here are similar to that found by Feiner et al. (2016) in an Alabama forest during SOAS (Southern Oxidant and Aerosol Study) where isoprene was the dominant BVOC. In that study, the modeled OH concentrations using MCM 3.3.1 were in good agreement with the measured concentrations when interferences were subtracted from the measurements.

While the predictions by both the RACM and MCM models are within the combined uncertainty of the measurements and the models, the MCM 3.3.1 results are in better agreement with the measurements (Fig. 4), which could suggest that the measurements are consistent with the lower effective bulk 1,6-H shift peroxy radical isomerization rate constants in MCM 3.3.1 compared to the original LIM1 recommended rates (Novelli et al., 2020). This is in contrast to the results of from the IRRONIC campaign discussed above, where the MCM 3.3.1 model underpredicted the measured concentrations by approximately a factor of 2, with the RACM2-LIM1 model predictions in better agreement with the measurements (Lew et al., 2020). Similarly, Novelli et al. (2020) reported that the MCM 3.3.1 mechanism underpredicted measurements of OH by a factor of approximately 1.4 during isoprene oxidation experiments in the SAPHIR (Simulation of Atmospheric Photochemistry In a large Reaction) chamber when mixing ratios of NO were less than 0.2 ppb. Unfortunately, the poor precision of the OH measurements reported here do not allow a robust test of the two mechanisms.

### 3.3 HO$_2$ and XO$_2$ measurements and model predictions

The time series of measured and modelled HO$_2$, and total peroxy radical (XO$_2$) concentrations from July 2 through July 31 are shown in Fig. 3 and correlation plots of the measured concentrations and MCM 3.3.1 model predictions are shown in Fig. S2. Daily maxima were typically observed between 13:30 and 15:30 local time and ranged from 6.7 ppt ($1.7 \times 10^8$ cm$^{-3}$) to 28.2 ppt ($7.0 \times 10^8$ cm$^{-3}$) for HO$_2$ and 10.8 ppt ($2.7 \times 10^8$ cm$^{-3}$) to 52.1 ppt ($1.3 \times 10^9$ cm$^{-3}$) for XO$_2$. Diurnal average profiles are shown in Fig. 4, in addition to the model results. The average maximum (12:00–15:00) of HO$_2$ measurements performed during the campaign was 11.6 ppt ($2.85 \times 10^8$ cm$^{-3}$), while the maximum daytime average of the XO$_2$ measurements was approximately 29.0 ppt ($7.7 \times 10^8$ cm$^{-3}$).

The measured HO$_2$ concentrations were similar to previous measurements at this site. Median daytime maximum concentrations of HO$_2$* measured in 2008 were approximately 28 ppt ($7 \times 10^8$ cm$^{-3}$), while median daytime maximum concentrations of HO$_2$* measured in 2009 were approximately 20 ppt ($5 \times 10^8$ cm$^{-3}$) (Griffith et al., 2013). The conversion efficiency of isoprene peroxy radicals to the measured HO$_2$* concentrations during these studies was estimated to be approximately 90%, suggesting that the measured HO$_2$* concentrations reflected the sum of HO$_2$ + isoprene peroxy radicals. Given that isoprene peroxy radicals contribute to approximately 33% of the total peroxy radical concentrations, the measured HO$_2$* concentrations in 2008 and 2009 were greater than HO$_2$ but less than XO$_2$ concentrations. When compared to 2009, the higher HO$_2$* concentrations measured in 2008 were likely due to the higher mixing ratios of HCHO observed in 2008 leading to greater rates of radical production (Griffith et al., 2013). The higher mixing ratios of HCHO may be a result of the higher mixing ratios of isoprene leading to a greater production of HCHO during the warmer temperatures observed in 2008 (Griffith et al., 2013).

Average daytime maximum concentrations of HO$_2$ measured at this site in 1998 were reported to be approximately 16 ppt ($3.9 \times 10^8$ cm$^{-3}$) (Tan et al., 2001), similar to that measured in this study. However, it is not clear whether the 1998 HO$_2$



measurements were influenced by interferences from isoprene-based peroxy radicals as discussed above. As a result, these
measurements may be an upper limit to the actual $HO_2$ concentrations. The measured $XO_2$ concentrations are similar to the total
$RO_2 + HO_2$ concentrations measured at this site in 1997 by Mihele and Hastie (2003), who reported daytime maximum mixing ratios
between 20 and 65 ppt using a radical chemical amplifier technique.

As illustrated in Fig. 4, the base RACM2 and MCM 3.2 models overpredict both the measured $HO_2$ and $XO_2$
concentrations during the daytime, although the agreement with the measured $HO_2$ concentrations is within the combined
uncertainty of the measurements and the model. The base RACM2 model overpredicts the measured average maximum $HO_2$
concentrations by approximately 30%, while the MCM 3.2 overpredicts the measured daytime maximum $HO_2$ by approximately
60%. However, including the LIM1 isoprene oxidation mechanism increases the daytime $HO_2$ concentrations predicted by the base
models by approximately 15% and 35% for MCM and RACM2 models, respectively (Fig. 4). Overall, the both the RACM2-LIM1
and MCM 3.3.1 models overpredict the measured daytime maximum $HO_2$ concentrations by approximately 80% which is outside
of the combined measurement and model uncertainties. Similarly, the base RACM2 and MCM 3.2 models as well as the updated
RACM2-LIM1 and the MCM 3.3.1 models overpredict the daytime $XO_2$ concentrations by more than a factor of 2, with predicted
daytime maximum $XO_2$ mixing ratios ranging from 65.5 (RACM2) to 72.6 (RACM2-LIM1) ppt ($1.6 - 1.8 \times 10^9$ cm$^{-3}$).

The model overprediction of the daytime measured $HO_2$ concentrations is consistent with model simulations of the
measured $HO_2^*$ concentrations at this site in 2008 and 2009 (Griffith et al., 2013). In 2008, a base RACM model overpredicted the
measured $HO_2^*$ concentrations by approximately 30% on average, while the same model overpredicted the $HO_2^*$ concentrations
measured in 2009 by approximately 50%. Similar to the results presented here, addition of the LIM1 mechanism for isoprene
oxidation to the RACM model likely would have increased the discrepancy between the 2008 and 2009 measurements. However,
these model results are in contrast to that observed at this site in 1998, where a RACM-based model was able to reproduce the
reported measured $HO_2$ concentrations (Tan et al., 2001). As discussed above, these measurements likely represent an upper limit
to the actual $HO_2$ concentrations as it is not clear whether the measurements of $HO_2$ were free from interferences from isoprene-
based and other alkene-based peroxy radicals (Fuchs et al., 2011; Lew et al., 2018). As a result, it is likely that this RACM-based
model of Tan et al. (2001) overestimated the actual $HO_2$ concentrations in 1998. The results reported here are also in contrast to
the results of Mihele and Hastie (2003), who found that a 0-D model using the MCM chemical mechanism could reproduce the
measured daytime $XO_2$ concentrations on several days. Similarly, these results are in contrast to the IRRONIC campaign, where
the MCM and RACM2 models were able to reproduce the measured $HO_2^*$ concentrations to within 30% (Lew et al., 2020), as
well as the results from SOAS, where the MCM 3.3.1 model was able to reproduce the measured $HO_2$ concentrations to within the
combined uncertainties of the measurement and the model (Feiner et al., 2016). The ability of the models to reproduce the measured
peroxy radical concentrations in these studies may reflect the higher mixing ratios of $NO_x$ observed at the PROPHET site in 1997,
the SOAS site, and at the IRRONIC site, resulting in a greater contribution of the $RO_2 + NO_x$ reactions to the fate of peroxy radicals
during these campaigns (Mihele and Hastie, 2003; Sanchez et al., 2018; Lew et al., 2020). An analysis of the discrepancies between
the PROPHET and IRRONIC campaigns will be presented in a subsequent paper.

The composition of the total peroxy radical concentration ($XO_2$) in the RACM2-LIM1 model is shown in Fig. 5. As
illustrated in this figure, the model predicts that $HO_2$ radicals comprise approximately 33% of the total daytime maximum $XO_2$
concentration, with isoprene peroxy radicals accounting for approximately 33%, methyl peroxy radicals approximately 12%, acyl
peroxy radicals approximately 7%, and peroxy radicals from alkane, alkene, and terpene oxidation comprising the remaining 15%.
Given that the model agreement with the measurements is better for $HO_2$, the majority of the discrepancy between the modeled
and measured $XO_2$ concentrations is likely due to a greater overestimation of $RO_2$ radicals, including isoprene-based peroxy
radicals. These results are in contrast to that reported by Kundu et al. (2019), who found that their measurements of $XO_2$



concentrations by the ECHAMP instrument during the IRRONIC campaign could be reproduced on several days by a model
incorporating the MCM 3.2 chemical mechanism. As discussed above, the ability of the models to reproduce the measured concentrations during IRRONIC may reflect the higher mixing ratios of NO observed during this campaign, resulting in a greater contribution of $RO_2 + NO_x$ reactions to the fate of peroxy radicals at this site.

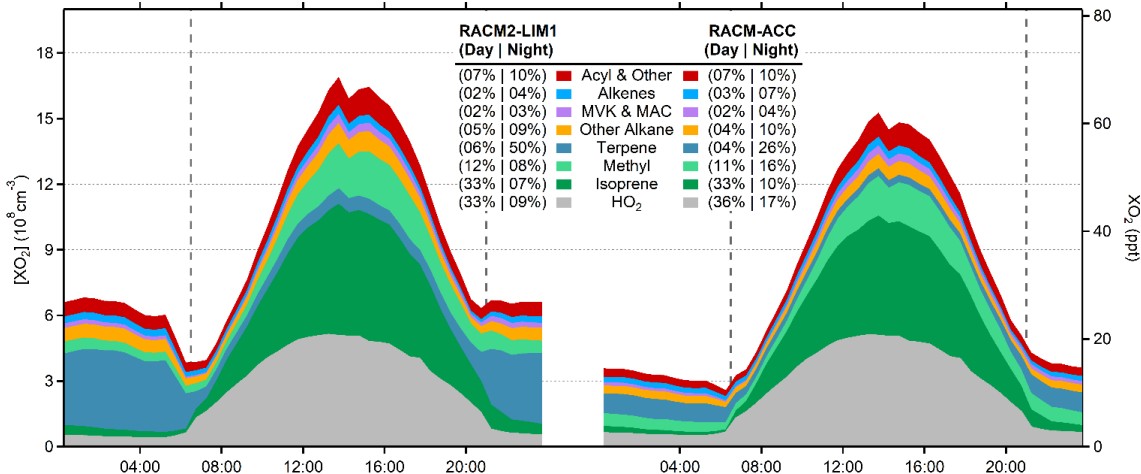

**Figure 5: Modeled XO₂ composition from RACM2-LIM1 (left) and RACM-ACC (right). Colors represent peroxy radicals derived from**
**the listed VOCs and numbers represent the percentage contribution of each species to the total concentration of XO₂ during the day (06:30 to 21:00) and at night (21:00 to 06:30), respectively.**

During the nighttime, the models reproduce the measured $HO_2$ concentrations but overestimate the measured $XO_2$ radical concentrations (Fig. 4). The RACM2 and MCM models overpredict the nighttime $XO_2$ concentrations by factor of approximately 4, with the RACM2-LIM1 model predicting mixing ratios of total peroxy radicals of approximately 27 ppt between 21:00 and 6:00
and the MCM 3.3.1 model predicting $XO_2$ mixing ratios of approximately 36 ppt during the night (Fig. 4) compared to the measured concentrations of less than 10 ppt. This is in contrast to the results of Mihele and Hastie (2003), who found that the MCM could reproduce the nighttime measured $XO_2$ mixing ratios that were generally below 10 ppt in their 0-D model, as well as the results of Kundu et al. (2019), who also found that the MCM 3.2 chemical mechanism could reproduce the measured nighttime mixing ratios of less than 10 ppt during the IRRONIC campaign.

The RACM-LIM1 model predicts that approximately 50% of the nighttime total $XO_2$ radical concentration is composed of peroxy radicals derived from the ozonolysis of monoterpenes (Fig. 5). As mentioned above, the measured sum of monoterpenes was constrained as α-pinene in all model simulations, resulting in an average monoterpene ozonolysis rate constant that is likely similar to that expected from previous speciated measurements of monoterpenes, including limonene and β-pinene, at this site (Ortega et al., 2007; Kim et al., 2011). However, the average ozonolysis rate constant assumed in the model could represent an
upper limit if the monoterpene composition was dominated by species less reactive with ozone (e.g. camphene, cymene) or a lower limit if more reactive terpene species were present (e.g. ocimene, limonene) (Atkinson et al., 1990; Khamaganov and Hites, 2001; Atkinson and Arey, 2003).

The addition of the $RO_2 + RO_2$ accretion reactions described above to the RACM-LIM1 and MCM 3.3.1 models (RACM-ACC and MCM-ACC) significantly reduces the predicted $XO_2$ radical concentrations at night by 50%, lowering the
measurement/model discrepancy to less than 5 ppt for the RACM-ACC model. As shown in Fig. 5, this is largely due to a reduction in the concentration of organic peroxy radicals derived from monoterpenes due to the relatively large rate constants for the associated $RO_2 + RO_2$ accretion reactions (Table 2).



**3.4 Radical budget analysis**

A radical budget analysis for OH, HO$_2$, isoprene-based peroxy radicals (ISOP) and total RO$_x$ was conducted to provide

information about the processes that drive radical production and the radical loss pathways in this environment and also to highlight

the relative importance of the changes in radical chemistry upon the addition of the LIM1 mechanism and accretion reactions.

Figure 6a illustrates the campaign average production and loss pathways of OH for the RACM2-LIM1 model. This includes both

initiation reactions and propagation steps that produce OH in blue, while termination pathways are shown alongside propagation

steps that convert OH to HO$_2$ or RO$_2$ in red. The addition of the LIM1 reactions increases the maximum OH production rate at

13:45 by 35% from 2.01 ppb h$^{-1}$ in RACM2, to 2.71 ppb h$^{-1}$ in RACM2-LIM1, primarily due to the isomerization of isoprene

peroxy radicals to form HPALDs, which readily photolyze to form OH, and also di-HPCARP radicals, which rapidly decompose

to produce additional OH radicals (Peeters et al., 2014; Teng et al., 2017; Wennberg et al., 2018). In the morning (6:45-13:15),

RACM2-LIM1 suggests HO$_2$ + NO reaction is the dominant source of OH radicals, accounting for 41% of total OH production,

and as much as 53% when NO mixing ratios are highest. This decreases to 21% in the afternoon and evening as the NO

concentration decreases. Photolytic processes are significant throughout the day, with ozone and HONO photolysis contributing

up to 28% and 13% respectively during the day. Ozonolysis of alkenes, primarily monoterpenes, is a minor contributor of up to

6% during the day but is the dominant source of OH at night.

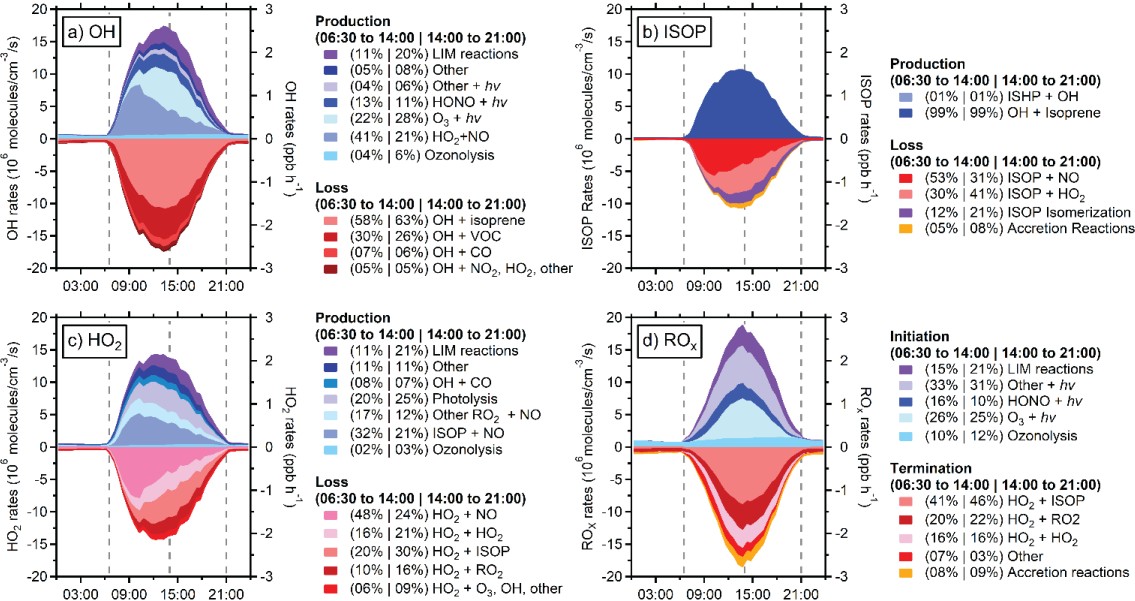

**Figure 6: Radical budgets from the RACM2-LIM1 model with additional accretion reactions (RACM-ACC) for a) OH, b) isoprene-**
**based peroxy radicals (ISOP), C) HO$_2$, and d) total ROx. Shades of blue represent reactions that produce/initiate radicals, and shades of red represent radical loss/termination reactions. LIM reactions (purple) include reactions added as part of the Leuven Isoprene Mechanism. Percentages represent the relative initiation or termination rates of each respective process in the morning (06:30: to 14:00) and during the evening (14:00 to 21:00) which are indicated by the vertical dashed lines.**

Reaction with isoprene is the dominant loss pathway for OH radicals accounting for approximately 60% of daytime OH

reactivity. Other VOCs (16%), namely monoterpenes, and OVOCs (10%), such as formaldehyde, methyl vinyl ketone, and

methacrolein, make up the majority of the remaining daytime OH reactivity. Propagation through reaction with CO is minor (6%),

and termination through the OH + NO$_2$ reaction is not significant (<2%). Consistent with the OH radical budget, the OH + isoprene

reaction is the dominant source of isoprene-based peroxy radicals (ISOP; Fig. 6b), with the ISOP + NO reaction accounting for



approximately 53% of their total loss in the morning, while the ISOP + HO$_2$ reaction and peroxy radical isomerization reactions in the LIM1 mechanism account for 62% of isoprene-based peroxy radical loss in the afternoon. The ISOP accretion reaction accounts for only 8% of the loss of isoprene-based peroxy radicals in the afternoon.

Figure 6c illustrates the campaign average HO$_2$ radical production and loss pathways for RACM2-LIM1. The production of HO$_2$ in the RACM-LIM1 model is largely due to turnover from the RO$_2$ + NO reaction. During the morning, when NO concentrations are greatest, 32% of HO$_2$ is produced from the reaction of NO and peroxy radicals derived from isoprene while 17% is produced from the reaction of NO with other RO$_2$ species. In addition to reactions with NO, the photolysis of formaldehyde and other carbonyls can account for more than 20% of daytime HO$_2$ production, and turnover from the OH + CO reaction near 8%. HO$_2$ loss is primarily due to reaction with NO in the morning (48%) but dominated by the HO$_2$ self-reaction, reaction with isoprene RO$_2$ to form ISOPOOH, and reaction with other peroxy radicals in the afternoon and evening (21%, 30%, and 16% respectively).

The total RO$_x$ radical budget is illustrated in Fig. 6d. The addition of LIM1 reactions increases the maximum radical initiation rate by 28% from 2.11 to 2.69 ppb h$^{-1}$, again primarily due to fast photolysis of HPALDs and decomposition of di-HPCARP radicals. Overall radical initiation in RACM2-LIM1 is largely due to photolytic processes, with a combined 51% from ozone photolysis (26%), HONO (13%), and HPALDs (18%), and 32% from the photolysis of other species such as hydrogen peroxide, aldehydes, organic peroxides, and nitric acid. Ozonolysis is a consistent radical initiation source of approximately 0.21 ppb h$^{-1}$ throughout the day, which dominates RO$_x$ initiation at night and is a significant contributor (11%) throughout the day when photolysis sources are dominant.

Daytime termination of radicals in RACM2-LIM1 is dominated by peroxy radical self- and cross-reactions, primarily the reaction of isoprene peroxy radicals with HO$_2$ to form ISOPOOH (44%) but also the HO$_2$ self-reaction (16%) and HO$_2$ + other RO$_2$ species (21%). Radical reactions with NO$_x$ were less significant due to the low NO$_x$ concentrations and accounted for at most 0.03 ppb h$^{-1}$, or less than 2%, of the RACM-LIM1 termination budget. The addition of RO$_2$ + RO$_2$ accretion reactions in the RACM-ACC model provides an alternative pathway that results in a termination rate equivalent to half that of HO$_2$ + RO$_2$ reactions at night (4.0 × 10$^5$ molecules cm$^{-3}$ s$^{-1}$) and accounts for 30% of total RO$_x$ termination during this time. During the day, when isoprene and NO mixing ratios are higher, these reactions only contribute to 9% of the overall termination due to the lower rate constants for reactions of C$_5$-RO$_2$ from isoprene (Table 2). As shown in Fig. 4, this results in better agreement between the measurement and model at night, but model overprediction during the day remains.

## 4 Discussion

As illustrated in Fig. 4, including the accretion reactions shown in Table 2 into both the RACM2-LIM1 and MCM 3.3.1 models improves the agreement between the model and measured XO$_2$ concentrations at night to within the combined uncertainty of the model and the measurements, although the agreement of the RACM2 model is better. However, including these accretion reactions in the model only decreases the modeled XO$_2$ concentrations by 9% during the daytime when isoprene-based peroxy radicals dominate the total XO$_2$ composition (Fig. 5), as the RO$_2$ + RO$_2$ accretion rate constants for isoprene-based peroxy radicals are smaller compared to that for monoterpene-based peroxy radicals (Table 2).

One possible explanation for the model discrepancies with the measured HO$_2$ and XO$_2$ concentrations during the daytime are errors associated with the measurements of these radicals, such as a systematic error in the calibration of HO$_2$ or XO$_2$ radicals. However, as discussed above, measurements of XO$_2$ concentrations during the IRRONIC campaign were in good agreement with model predictions by the RACM2 and MCM mechanisms, where isoprene dominated OH reactivity during the daytime and



isoprene-based peroxy radicals likely contributed to approximately 30% of the total $XO_2$ concentrations, similar to that observed during PROPHET-AMOS (Kundu et al., 2019). While measurements of $HO_2$ were not conducted during IRRONIC, the measured $HO_2^*$ concentrations were also found to be in good agreement with the model predictions (Lew et al., 2020). In addition, the

measured $XO_2/HO_2^*$ ratio was found to be in good agreement with the modelled ratio (Kundu et al., 2019). While these results do not rule out the possibility of errors associated with the calibration of the ECHAMP and IU-FAGE instruments, they suggest that the discrepancy between the measurement and model predictions during PROHET-AMOS may not be due to a systematic error in the measurements. As noted in Sect. 2.3, ECHAMP is expected to be 8% less sensitive to isoprene $RO_2$ than other peroxy radicals. As the modeled isoprene $RO_2$ mixing ratio accounted for approximately 33% of modeled $XO_2$ during the daytime (Fig. 5), this

suggests that the measured $XO_2$ represents a lower limit and could be as much as 3% higher than reported. Given the large differences between modeled and measured $XO_2$ of more than a factor of two during mid-day, this difference cannot account for the discrepancy with the modeled concentrations.

        Measurements of isoprene hydroxy hydroperoxides (ISOPOOH) produced from the reaction of isoprene-based $RO_2$ radicals with $HO_2$ can provide an additional test of the model chemistry at this site. Figure 7 shows the average ISOPOOH mixing

ratio measured during PROPHET-AMOS between July 22 and July 27 by the Caltech low-pressure GC-CIMS instrument (Vasquez et al., 2018). The measurements shown include both the 1,2- and 3,4-ISOPOOH isomers, although the 1,2-ISOPOOH constitutes the dominant fraction (Vasquez et al., 2018). In order to achieve a more realistic comparison, a measurement-based deposition term for ISOPOOH only (Nguyen et al., 2015; Wei et al., 2021) was included in the mechanism for all model runs shown in this figure. Still, as illustrated in Fig. 7, the RACM2-ACC and MCM-ACC models overpredict the measured ISOPOOH concentrations

by approximately a factor of 8-10 during the daytime, consistent with the overprediction of peroxy radicals by the models. Constraining the model to the measured concentrations of $HO_2$ and isoprene-$RO_2$ (assuming the same relative distribution of $RO_2$ radicals predicted by the models) improves the agreement (Fig. 7). This suggests that the measured $HO_2$ and $XO_2$ concentrations are consistent with the measured ISOPOOH concentration and that the models are overpredicting the concentrations of $HO_2$ and isoprene-based peroxy radicals, either through an overestimation of their production or an underestimation of their loss.

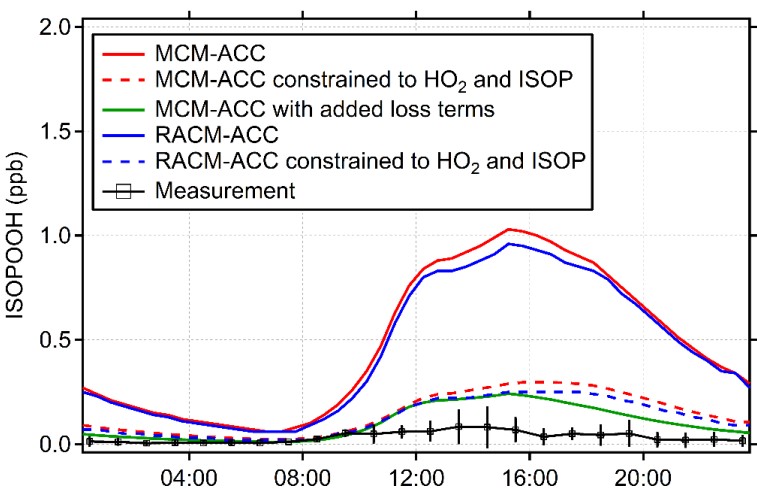


**Figure 7: Measured and modelled isoprene hydroxy hydroperoxide (ISOPOOH) mixing ratios. Measurements are an average from July 23-27 (Vasquez et al., 2018). The solid lines represent modeled mixing ratios from the RACM-ACC and MCM-ACC models, the dashed lines represent predictions of the same model constrained to measured values of HO₂ and measurements of XO₂ scaled to the modeled isoprene RO₂ composition.**





The radical budget analysis suggests that the OH + isoprene reaction is the main source of isoprene-based peroxy radicals during PROPHET-AMOS (Fig. 6b). Measurements of the total OH reactivity together with the measurement of the concentration of OH can provide an estimate of the rate of peroxy radical production from reactions of VOCs with OH. Measurements of total OH reactivity were also conducted during PROPHET-AMOS using both the Indiana University Total OH Loss Method (IU-TOHLM) instrument (Hansen et al., 2014) and the IMT Nord Europe Comparative Reactivity Measurement (CRM) instrument

(Hansen et al., 2015), and an analysis of the results and the instrument intercomparison will be presented in a subsequent paper. Figure 8 shows the diurnal averaged total OH reactivity as measured by the IU-TOHLM instrument along with that predicted by the RACM2-LIM1 model. As illustrated in this figure, the measured OH reactivity agreed with that calculated from measured and modeled OH sinks, including the reactivity of some unmeasured oxidation products, suggesting that the loss of OH is well represented by the models. Reaction with isoprene is the dominant daytime OH radical sink, accounting for approximately 60% of

the total OH reactivity during the day, in both the RACM2-LIM1 (Fig. 8) and MCM 3.3.1 (Fig. S4) models.

The reasonable agreement between the measured and modeled OH concentrations and total OH reactivity suggests that the rate of production of peroxy radicals by the reaction of OH with isoprene and other VOCs is not overestimated by the model given that these reactions are the dominant source of peroxy radicals during PROPHET-AMOS. In addition, because radical propagation by the $RO_2$ + NO reaction is a major source of $HO_2$ radicals, it is unlikely that the model is overestimating the

production of $HO_2$ radicals, although photolysis of HCHO and other aldehydes are also predicted to be a significant source of $HO_2$ radicals, contributing up to 20-25% of total $HO_2$ production. However, HCHO and other aldehydes were measured during the campaign, providing a constraint on radical production by the photolysis of these compounds.

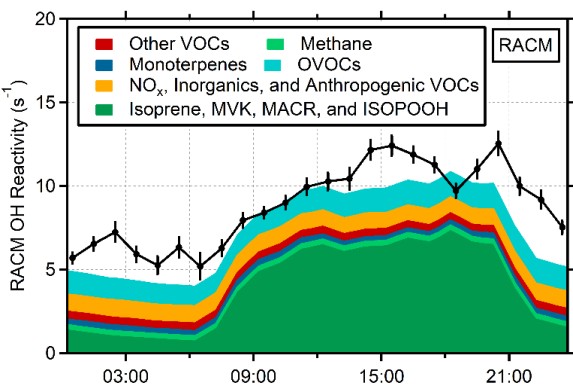

**Figure 8: Diurnal average of the measured (IU-TOHLM instrument) and modeled total OH reactivity at the top of the tower during**
**PROPHET-AMOS. Modeled reactivity is largely based on measured species that are used as constraints in the model but also includes contributions from unmeasured oxidation products in the RACM-LIM1 model.**

Reactant segregation between OH radicals and isoprene could lead to an effective reduction in the rate of this reaction, resulting in an overestimation of the reaction rate by the models. While it has been suggested that segregation between OH and isoprene could effectively reduce the rate of the OH + isoprene reaction by 60% (Butler et al., 2008), recent studies have suggested

that segregation of OH and isoprene may result in an effective reduction in the rate of the OH + isoprene reaction of less than 15% (Ouwersloot et al., 2011; Pugh et al., 2011). As a result, it is unlikely that reactant segregation is responsible for the discrepancy between the measured and modeled $HO_2$ and $XO_2$ concentrations described above, and the overprediction of these peroxy radicals by the models is likely due to an underestimation of radical termination rather than an overestimation of the production of these radicals. An additional loss of $HO_2$ and isoprene-based $RO_2$ radicals on the order of the rate of these peroxy radicals with NO is

needed in order to resolve the daytime discrepancy between the model and the measurements. Figure 4 includes the results of an





additional model that features the RACM2-ACC chemical mechanism but also includes additional sinks for $HO_2$ and isoprene peroxy radicals (green line in Fig. 4). The added $HO_2$ sink corresponds to a first-order loss rate of 0.012 $s^{-1}$, which is approximately 40% of the daytime $HO_2$ loss, while the added isoprene-based $RO_2$ sink corresponds to a first-order loss rate of 0.024 $s^{-1}$, which is approximately 60% of the daytime loss of isoprene-based peroxy radicals (Fig. S3). The addition of these peroxy radical loss

mechanisms reduces the predicted daytime maximum OH concentrations by 25% (Fig. 4a). These loss processes could potentially include several components, such as uptake of radicals and important precursors to aerosols or the forest canopy, faster self- and cross-reactions between $C_5$-$RO_2$ and other $RO_2$ species that serve as $RO_2$ radical sinks, or reaction of $RO_2$ radicals with isoprene or other unsaturated VOCs.

The first-order loss of $HO_2$ on aerosols can be estimated assuming a first-order loss to aerosol surfaces (Ravishankara,

1997; Whalley et al., 2010) (Eq. 1) where $A$ is the aerosol surface area per volume ($cm^2$ $cm^{-3}$), $\gamma$ is the uptake coefficient, $c_g$ is the mean molecular speed of a gas (cm $s^{-1}$) given by Eq. 2 where $R$ is the gas constant, $T$ is the temperature and $M_w$ is the molecular weight of the gas. Aerosol uptake coefficients for $HO_2$ radicals have been measured in both laboratory and field studies, with values ranging from less than 0.1 to 0.4 (Taketani et al., 2008; Thornton et al., 2008; Taketani et al., 2012; George et al., 2013; Zhou et al., 2021). Assuming values of $A = 100$ $\mu m^2$ $cm^{-3}$ typical of rural aerosols (Cai et al., 2017) and $\gamma = 0.1$ results in an

estimated first-order loss rate of approximately 0.001 $s^{-1}$, while assuming values of $A = 200$ $\mu m^2$ $cm^{-3}$ and $\gamma = 0.4$ results in an estimated first-order loss of approximately 0.008 $s^{-1}$. Similar assumptions for isoprene-based peroxy radicals result in an estimated first-order loss of approximately $6 \times 10^{-4} - 5 \times 10^{-3}$ $s^{-1}$. Assuming an uptake coefficient of $\gamma = 1$ for isoprene-based peroxy radicals would lead to estimated first-order loss rates of approximately $6\times10^{-3} - 1 \times 10^{-2}$ $s^{-1}$. These results suggest that while heterogeneous loss of peroxy radicals on aerosols may contribute to the model overestimation of the measurements they may not be the only loss

mechanism missing in the model.

$$k'_{loss} = \frac{c_g A \gamma}{4}, \tag{2}$$

$$c_g = \sqrt{\frac{8RT}{\pi M_w}}, \tag{3}$$

Recent studies have detected products of the reaction of $RO_2$ radicals with unsaturated VOCs under atmospheric conditions and suggested that the reaction of isoprene-based peroxy radicals with isoprene could be a significant radical termination

reaction in low $NO_x$ regions (NO ≤ 0.05 ppb) (Noziere et al., 2023). Assuming a rate constant of $10^{-14}$ $cm^3$ $s^{-1}$ for this reaction based on measurements of the rate of the reaction of acyl peroxy radicals with 2,3-dimethyl-2-butene, the reaction of isoprene-based peroxy radicals with isoprene would result in an estimated first-order loss of approximately $5 \times 10^{-4}$ $s^{-1}$. Although this $RO_2$ + alkene rate coefficient is not large enough to resolve the discrepancy between the measured and modeled $XO_2$ mixing ratios at the PROPHET site, $RO_2$ radicals derived from the OH-oxidation of isoprene and monoterpenes could exhibit enhanced reactivity

to alkenes and constitute a more significant portion of the missing radical sink (Nozière and Fache, 2021).

An underestimation of radical termination by the reactions of isoprene-based $RO_2$ radicals could also be responsible for the discrepancies between the modeled and measured peroxy radical concentrations. The overestimation of the measured ISOPOOH concentration by the model (Fig. 10) suggest that the model is not underestimating the rate of radical termination by the reaction of $HO_2$ with isoprene-based $RO_2$ radicals. To account for the missing loss of isoprene-based $RO_2$ radicals, an accretion

rate constant for the self-reaction of isoprene-based $RO_2$ radicals of approximately $4 \times 10^{-11}$ $cm^3$ $molecule^{-1}$ $s^{-1}$, similar to that for the self-reaction of monoterpene $RO_2$ radicals (Table 2), would bring the modeled peroxy radical concentrations into agreement with the measurements. While this is greater than the factor of 2-3 uncertainty associated with the measured rate constant for this reaction (Berndt et al., 2018), a combination of loss rates from aerosol uptake, $RO_2$ reactions with alkenes, and accretion reactions



would require a smaller accretion rate constant for isoprene-based $RO_2$ radicals. An analysis of the experimental radical budgets
including the impact of potential additional loss rates will be presented in a subsequent publication.

Another potential loss process in the 0-D model includes vertical and/or horizontal transport of peroxy radicals given their relatively longer modeled lifetimes under the low $NO_x$ conditions at the PROPHET site. The average chemical lifetimes of $HO_2$ and isoprene-based peroxy radicals during the daytime range from 35–135 s and 40–160 s, respectively. These calculated lifetimes depend primarily on the reactions of $HO_2$ and isoprene-based $RO_2$ with the measured radical concentrations and the measured
concentration of NO, but also on the reactions of $HO_2$ with $O_3$ and the isoprene $RO_2$ isomerization reactions included in the LIM1 mechanism. These lifetimes are on the order of the expected canopy mixing timescale in forested environments (~2 min) (Wolfe et al., 2011; Wei et al., 2021), suggesting that deposition to the canopy surface could constitute a portion of the missing radical loss process, and could be more significant on well-mixed days.

## 5 Summary

The daytime maximum measured OH radical concentrations during the PROPHET-AMOS campaign were generally in good agreement with model simulations using both the RACM2 and MCM 3.2 chemical mechanisms, though both models overestimated the measured values in the morning. In contrast to previous measurements by the IU-FAGE instrument, no significant OH interferences were measured during the campaign, perhaps due to the lower temperatures and ozone concentrations, which seem to be correlated with unknown interferences associated with the LIF-FAGE technique (Lew et al., 2020). Including the LIM1
isoprene chemical mechanism into the RACM2-LIM1 and MCM 3.3.1 models increases the maximum modeled OH concentration by approximately 30%, with the MCM 3.3.1 mechanism in better agreement with the measurements. These results are in contrast to previous measurements in forest environments, where the measurements were found to be significantly greater than model predictions (Rohrer et al., 2014).

Both the RACM2 and MCM models overpredict the measured daytime concentration of $HO_2$ by approximately 50% and
the measured $XO_2$ concentrations by approximately a factor of 2, similar to previous measurements at this site (Griffith et al., 2013). During the nighttime, the models are able to reproduce the measured $HO_2$ concentrations but overestimate the measured $XO_2$ radical concentrations by factors of approximately 3-5, with approximately 50% of the nighttime total $XO_2$ radical concentration composed of peroxy radicals derived from the ozonolysis of monoterpenes. The addition of the $RO_2 + RO_2$ accretion reactions to the models significantly reduces the predicted $XO_2$ radical concentrations at night by up to 60% due to the relatively
large rate constants for the $RO_2 + RO_2$ accretion reactions of monoterpene-derived peroxy radicals. However, including these $RO_2 + RO_2$ accretion reactions does not significantly impact the modeled daytime peroxy radical concentrations when isoprene-based peroxy radicals dominate the total $XO_2$ composition, as the reported $RO_2 + RO_2$ accretion rate constants for isoprene-based peroxy radicals are smaller compared to that for monoterpene-based peroxy radicals.

The models also overpredict the daytime measured concentrations of isoprene hydroxy hydroperoxide, consistent with an
overprediction of the concentration of isoprene-based peroxy radicals. Constraining the model to the measured peroxy radical concentrations improves the agreement with the measured ISOPOOH concentrations. These results suggest that the measured radical concentrations are more consistent with the measured ISOPOOH concentrations, providing additional confidence in the accuracy of the $HO_2$ and $XO_2$ radical measurements, and suggest that the model is either overestimating the production of peroxy radicals or underestimating their loss. The modeled OH concentrations and total OH reactivity were in good agreement with the
measurements, suggesting that the model is not overestimating the production of peroxy radicals, including isoprene-based peroxy radicals.



To reproduce the measured peroxy radical concentrations, an additional loss process equivalent to the reaction of peroxy radicals with NO must be added to the model, accounting for approximately 60% of the total rate of radical termination in the model. The additional loss processes could potentially include several components, such as direct surface deposition of radicals

and important precursors to aerosols or the forest canopy, faster self- and cross- reactions between $C_5$-$RO_2$ and other $RO_2$ species, reactions of peroxy radicals with isoprene and other alkenes, or vertical and horizontal transport of peroxy radicals given their longer lifetime under the low $NO_x$ conditions at the PROPHET site. The overestimation of peroxy radical concentrations suggests that current atmospheric chemistry models may be overestimating the rate of production of ozone and other secondary products in similar low $NO_x$ areas impacted by isoprene emissions. Additional measurements and modeling studies are needed to resolve these

discrepancies.

**Data availability.** Data presented in this study can be obtained from the authors upon request (pstevens@indiana.edu)

**Competing interests.** The authors declare that they have no conflicts of interest.

**Author contributions.** BB, ML, YW, PR, and PSS were responsible for the LIF-FAGE measurements of OH, $HO_2$, and OH reactivity. BD, MDR, DCA and EW were responsible for the ECHAMP measurements of $XO_2$. HDA and DBM were responsible for the PTR-MS measurements of VOCs and OVOCs. SD and TL were responsible for the GC measurements of VOCs and OVOCs. AW, GT, JO, and DM were responsible for the measurements of NO and $NO_2$. WW and DH were responsible for the measurements of $O_3$ at the Ameriflux tower. JF, ME, and SA were responsible for the measurements of photolysis frequencies and CO. JR, JS, and FK were responsible for the measurements of formaldehyde. HMA and JDC were responsible for the measurements of ISOPOOH. SB was responsible for coordination and preparation of the PROPHET site. BB, ML, YW, PR, and PSS conducted the analysis and photochemical modelling and wrote the paper with feedback from all co-authors.

**Acknowledgements.** This study was supported by the National Science Foundation, grants AGS-1440834 and AGS-1827450 to Indiana University, grant AGS-1443842 to the University of Massachusetts Amherst, grant AGS-1719918 to Drexel University, grant AGS-1561755 to the University of Colorado, grant AGS-1643306 to Harvard University, and grants AGS-1932771 and

AGS-1428257 to the University of Minnesota. We also thank Krystal Vasquez, Eric Praske, John Crounse, and Paul Wennberg for their hard effort obtaining the ISOPOOH measurements, Deedee Montzka for assistance in obtaining the $NO_x$ measurements, and all PROPHET-AMOS participants for making this work possible and the University of Michigan Biological Station for hosting the field study.



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
