# Peer review of "OH, HO2, and RO2 radical chemistry in a rural forest environment: Measurements, model comparisons, and evidence of a missing radical sink"

_EGUsphere, 2023_

## Author Comment (AC1)

We thank the reviewers for their insights and helpful suggestions. We feel that their feedback has led to changes that have strengthened the quality of the revised manuscript. Please find below the original reviewer comments in black, our responses in blue, and changes to the manuscript in red.

**Reviewer 1**

This study focuses on the comparison between measured OH, HO2 and the sum of HO2 and RO2 (XO2) radicals and results from 2 different chemical mechanisms, one the RACM (lumped) and the other the MCM (semi-explicit). Measurements were conducted in an isoprene dominated forest in Michigan where a lot of ancillary species as well as ISOPOOH were detected.

This study seems to suggest that quite a large loss rate (termination reaction) for both HO2 and XO2 is needed for both chemical mechanisms investigated to agree with the measurements. Several hypotheses are made such as fast deposition of the radicals on surfaces, faster than used RO2+RO2 reactions for isoprene RO2, RO2 radicals reaction with alkenes and/or segregation could play a role. The authors suggest that most probably a combination of all the above could explain the discrepancy although a rather large loss rate of about 60% during daytime is needed.

The paper is well written and structured, and the arguments are presented in a clear manner. The study is interesting but as it is not possible to give a clear conclusion on what is causing the discrepancies, I would recommend adding a bit of analysis to try and see if something more can be understood and after that I would recommend the publication.

My first suggestion would be to try and perform an experimental budget with the available data. I understand that it might not be possible for the HO2 and RO2 but it would be possible to perform it for the OH radical. In this way it should be clear if the very low OH observed in the morning hours is an instrument artefact as it is quite dubious. The OH budget would also help (possibly) to clarify if indeed there doesn't seem to be the need for additional sources of OH radicals as the comparison with the model seems to show. I have to say that isomerization reactions for isoprene-RO2 are more or less a given now so I am also wondering why there seems to be such a large overestimate of the OH radical when the most up to date mechs are used.

As suggested, we have added an experimental budget for OH to the supplement of the revised manuscript. As expected, the experimental budget is not balanced in the morning hours which suggests either a missing OH sink or an error with the measurement at this time. This is consistent with the discussion of the discrepancy between measured and modeled OH concentrations which was attributed to participant activity near the detection cell in the morning or a potential systematic measurement error.

An experimental OH budget based on measured concentrations of OH, $HO_2$, and other species, is also shown in Fig. S3. The imbalance between 7:00 and 12:00 suggests either a missing OH sink or errors with the OH measurement during this time.

[Figure]

**Figure S3:** Experimental OH radical budget. In panel (a), shades of blue represent reactions that produce OH, and shades of red represent loss rates, including reactions that propagate to $RO_2$ or $HO_2$. Percentages indicate the relative contribution of each respective process in the morning (06:30: to 14:00) and during the evening (14:00 to 21:00) time periods which are indicated by the vertical dashed lines. The net rate of production or loss is shown in panel (b).

The MCM v3.3.1 and RACM2-LIM1 models likely overestimate the OH concentrations due to a lack of adequate peroxy radical sinks that are relevant in this forested environment. This is illustrated by the green line in Figure 4, which represents a RACM-ACC model that features additional loss terms for $HO_2$ and isoprene-$RO_2$. The additional loss terms were discussed in terms of the necessary loss rate required to match the $HO_2$ and $XO_2$ measurements, but the impact of these loss terms on the modeled OH concentrations was only briefly mentioned – the following sentence has been added to page 23 of the revised manuscript to clarify.

The addition of these peroxy radical loss mechanisms reduces the predicted daytime maximum OH concentration by 25% to $1.65 \times 10^6$ $cm^{-3}$, which is within the combined uncertainties of the measurement and the model (Fig. 4a).

My second suggestion concerns the XRO2. I am wondering if it would not make sense to remove the HO2 fraction from it from the LIF measurement and then have a more or less RO2 measurement. I understand it does not make much of a difference since the measurement is compared with the some of HO2 and RO2 but to e able to compare with previous studies it would make it easier if it was RO2 instead of XO2.

We focused on comparing the direct measurements to their modeled counterparts (i.e. comparing $XO_2$ measurement to the modeled sum of $HO_2$ and $RO_2$) but do understand that this makes comparison to previous studies more complicated. As suggested by the reviewer we have added a plot that compares the average modeled $RO_2$ to the average measured $RO_2$ ($HO_2$ measured by LIF subtracted from $XO_2$ measured by ECHAMP) to the supplement of the revised manuscript.

In addition, measured $RO_2$ mixing ratios ($HO_2$ measured by LIF subtracted from $XO_2$ measured by ECHAMP) are compared with modeled $RO_2$ mixing ratios in Fig. S4.

[Figure]

**Figure S4:** Diurnal average measured (black) and modeled concentrations of $RO_2$. MCM models are shown in red and RACM2 in blue. The green line represents an additional version of the RACM-ACC model with added sinks for $HO_2$ and isoprene peroxy radicals. The measured $RO_2$ mixing ratios were determined by subtracting the measured $HO_2$ (LIF) from the measured $XO_2$ (ECHAMP).

My third suggestion is about the ISOPOOH. I am not aware of many ISOPOOH ambient measurements and although I could imagine a different publication focusing on that, it would be good to extend the discussion about it in this study. One thing that I find a bit odd is that the measured concentration of ISOPOOH is more or less zero (within the uncertainty) for the all time? I can see a bit of an increase but it is rather small. Even the model results after constraining HO2 and RO2 would expect quite a bit more. Could this be an instrument artefact? Is this consistent with previous measurements?

The reviewer is correct that there are not many measurements of ambient ISOPOOH available. The data in this manuscript were first presented in Vasquez et al. (2018). The measured mixing ratios for the sum of ISOPOOH isomers were between approximately 10-250 ppt, with an average maximum of approximately 100 ppt. We have added an inset in Figure 7a for better visibility. The measured mixing ratios of ISOPOOH were similar to that measured during the SOAS campaign (Kaiser et al., 2016). This has been clarified on pages 21 and 22 of the revised manuscript.

Measurements of isoprene hydroxy hydroperoxides (ISOPOOH) produced from the reaction of isoprene-based $RO_2$ radicals with $HO_2$ can provide an additional test of the model chemistry at this site. Figure 7 shows the average ISOPOOH mixing ratio measured during PROPHET-AMOS between July 22 and July 27 by the Caltech low-pressure GC-CIMS instrument (Vasquez et al., 2018). The measured mixing ratios were similar to that observed during the SOAS campaign (Kaiser et al., 2016).

[Figure]

**Figure 7:** Measured and modelled mixing ratios of (a) isoprene hydroxy hydroperoxides (ISOPOOH) and (b) isoprene hydroxy nitrates (IHN). Measurements of ISOPOOH are an average from July 23-27 (Vasquez et al., 2018) and measurements of IHN are an average of July 6-31. The solid lines represent modeled mixing ratios from MCM-ACC models, the dashed line represents predictions of the same model constrained to measured values of $HO_2$ and measurements of $XO_2$ scaled to the modeled isoprene $RO_2$ composition.

The modeled ISOPOOH is sensitive to the rate of ISOPOOH deposition. As described in the original manuscript, the model runs shown in Figure 7 include a literature-based ISOPOOH deposition term, but this term could vary significantly from forest to forest or with meteorological conditions. Due to this and also to the limited amount of ISOPOOH data Figure 7 is not intended to present a direct comparison between measured and modeled ISOPOOH, but instead to illustrate that the measured ISOPOOH is more consistent with the measured $HO_2$ and $XO_2$ radical concentrations, regardless of how ISOPOOH deposition is treated in the model. While the model constrained to the peroxy radical concentrations still overestimates the measurements, the model overestimation of the measured ISOPOOH in this study is similar to that observed during the SOAS campaign (Kaiser et al., 2016), where a large dilution rate was needed to bring the modeled ISOPOOH into agreement with the measurements. This has been clarified on page 21 of the revised manuscript.

Constraining the model to the measured concentrations of $HO_2$ and isoprene-$RO_2$ (assuming the same relative distribution of $RO_2$ radicals predicted by the models) improves the agreement (Fig. 7a), although the model still overestimates the measured concentrations. This overestimate of the measured ISOPOOH is similar to that observed during the SOAS campaign (Kaiser et al., 2016), where a large dilution rate was needed to bring the modeled ISOPOOH into agreement with the measurements.

Are there other products that show up which would compensate the production rate of RO2 that, as mentioned, was rather high, and the reacted isoprene must go somewhere.

In addition to ISOPOOH, measurements of isoprene hydroxy nitrates (IHN) produced from the reaction of isoprene peroxy radicals with NO were also overpredicted by the model. Constraining the model to the measured peroxy radical concentrations also brought the modeled IHN into

better agreement with the measurements, again suggesting that the measured isoprene hydroxy nitrates is more consistent with the measured peroxy radical concentrations. This result has been added to the revised manuscript (pages 21 and 22), and we have included a plot of the modeled and measured IHN in Figure 7b.

Similarly, the model also overestimates the concentrations of isoprene IHN produced from the reaction of isoprene peroxy radicals with NO and measured using iodine adduct CIMS (Xiong et al., 2015). Constraining the model to the measured peroxy radical concentrations improves the agreement with the measurements (Fig. 7b). It is also worth noting that the model does not account for losses of IHN due to reactive uptake onto aerosol and subsequent hydrolysis in the aerosol phase (Jacobs et al., 2014; Morales et al., 2021; Wang et al., 2021). Knowledge and incorporation of such loss rates in the model could better constrain the modeled IHN concentrations but the effect is expected to be small in comparison to the adjustment in the modeled output when constrained to measured $RO_2$. (Wei et al., 2021; Mayhew et al., 2022) These results suggest that the measured $HO_2$ and $XO_2$ concentrations are consistent with the measured ISOPOOH and IHN concentrations and that the models are overpredicting the concentrations of $HO_2$ and isoprene-based peroxy radicals, either through an overestimation of their production or an underestimation of their loss.

I noticed that on few occasions the subscripts are not correct so I would recommend checking and fixing that.

Thank you for noticing these mistakes. Subscript formatting errors have been corrected in the revised manuscript.

I also recommend adding the work by J. Medeiros et al. (2022) which is consistent with LIM1.

Thank you for the suggestion. We have added this reference to the introduction section discussing laboratory measurements of isoprene oxidation.

**References**

Kaiser, J., Skog, K. M., Baumann, K., Bertman, S. B., Brown, S. B., Brune, W. H., Crounse, J. D., de Gouw, J. A., Edgerton, E. S., Feiner, P. A., Goldstein, A. H., Koss, A., Misztal, P. K., Nguyen, T. B., Olson, K. F., St. Clair, J. M., Teng, A. P., Toma, S., Wennberg, P. O., Wild, R. J., Zhang, L., and Keutsch, F. N.: Speciation of OH reactivity above the canopy of an isoprene-dominated forest, Atmos. Chem. Phys., 16, 9349-9359, https://doi.org/10.5194/acp-16-9349-2016, 2016.

Vasquez, K. T., Allen, H. M., Crounse, J. D., Praske, E., Xu, L., Noelscher, A. C., and Wennberg, P. O.: Low-pressure gas chromatography with chemical ionization mass spectrometry for quantification of multifunctional organic compounds in the atmosphere, Atmos. Meas. Tech., 11, 6815-6832, https://doi.org/10.5194/amt-11-6815-2018, 2018.

---

## Author Comment (AC2)

We thank the reviewers for their insights and helpful suggestions. We feel that their feedback has led to changes that have strengthened the quality of the revised manuscript. Please find below the original reviewer comments in black, our responses in blue, and changes to the manuscript in red.

**Reviewer #2**

This paper presented the measurements of OH, HO2 and XO2 at the PROPHET site in July 2016. OH and HO2 were measured by FAGE with dedicated efforts to minimize the interference. XO2, the sum of HO2 and RO2, were measured by an ethane chemical amplification. The measured radical concentrations were consistent with previous field measurement in 2008 and 2009, considering the meteorological difference. The measurements were compared to box model calculations using RACM2-LIM1 and MCM3.3.1 as a standard test with updated information of isoprene oxidation mechanisms, which enhance the OH concentration by 30% and 20%, respectively, compared to those without LIM1 chemistry, namely RACM2 and MCM3.2. However, this standard models overpredicts OH, HO2 and XO2 more than 60%. RO2+RO2 accretion reactions were added to reduce the model-measurement discrepancy. Further radical budget analysis and model sensitivity tests indicated the model overprediction was related the missing radical sinks.

This paper is well-written and structured reasonable. The discussion is justified. I recommend the paper published in ACP after minor revision. Here is a few suggestions that might help to improve the paper.

1.  There were several radical measurements conducted at the PROPHET site. It would be very useful to show an overview plot to compare different field measurements. In selection 3.2 and 3.3, the measurements from 1998, 2008, 2009, and 2016 are compared in different position. For example, a bar plot summarizing the diurnal maximum OH, HO2, RO2 concentrations together with temperature, BVOCs concentrations would help the readers to interpretate the consistency/difference between several campaigns.

We thank the reviewer for this helpful suggestion. We have added Table S3 to concisely summarize the field campaigns conducted at the PROPHET site and added the following sentence to page 11 of the revised manuscript.

These measurements are summarized along with those from previous campaigns at the PROPHET site in Table S3

2.  As the authors suggested the radical sink related to surface loss may be relevant to the model overprediction, it would be interesting to have more quantitative analysis. One could calculate the probability of radical loss upon collision on the ground/canopy similar to the aerosol uptake. In this case, the surface area per volume can be derived from the ratio of ground/canopy area to the volume of mixing layer.

We agree with the reviewer that this section should be expanded, though studies investigating radical uptake on canopy surfaces are limited in comparison to studies involving $HO_2$ uptake to

aerosol surfaces. We have added a short paragraph on pages 24 and 25 of the revised manuscript that quantifies the uptake coefficients that are necessary to account for the missing $HO_2$ and isoprene $RO_2$ sink given a leaf area index of 3.8 $m^2$ $m^{-2}$ and assuming a mixing layer height of 1500 m.

Similar to the above discussion, radical loss to surfaces within the forest canopy can be estimated using Eq. 2 where $A$ now represents the ratio of the canopy surface area to the height of the mixing layer. Previous measurements at the PROPHET site reported a leaf area index (LAI) of approximately 3.8 $m^2$ $m^{-2}$ (Ortega et al., 2007). Assuming a mixing layer height of 1500 m, this suggests that an $HO_2$ uptake coefficient of $\gamma = 5 \times 10^{-4}$ would result in a first order loss rate of 0.013 $s^{-1}$, which could account for the proposed missing $HO_2$ sink. This uptake coefficient is lower than those measured for many atmospheric aerosols, but is similar to measurements of $HO_2$ uptake on organic aerosols (Lakey et al., 2015). Similarly, an uptake coefficient of $\gamma = 1.7 \times 10^{-3}$ for isoprene peroxy radicals would result in a first order loss rate of 0.024 $s^{-1}$ and could account for the missing radical sink. These results imply that loss to surfaces within the canopy could be a substantial radical loss mechanism in dense forests where low $NO_x$ mixing ratios result in longer peroxy radical lifetimes that are on the order of the transport time through the canopy.

Technical comments.

1. Please check the subscripts for HO2, RO2, XO2 are other chemicals throughout the paper.

Thank you for noticing these mistakes. We have corrected several subscript formatting errors in the revised manuscript.

2. The name of RACM2, RACM2-LIM1, MCM3.2, MCM3.3.1 should be used properly. For example, Line 565, it should be RACM2-LIM1 instead of RACM-LIM1.

As suggested, we have corrected the references to each model mechanism to be more consistent. In the revised manuscript RACM2 models are referred to as RACM2, RACM2-LIM1, or RACM-ACC, and MCM models are referred to as MCM v3.2, MCM v3.3.1, or MCM-ACC.

---

## Author Response (AR2)

Hi Lisa,

Thank you for the helpful suggestions. We agree that these simple changes have helped improve the manuscript. Please find below our responses in blue and specific changes to the manuscript in red.

**Editor Comments**

Dear Authors,

I have checked through the revised manuscript and your response to the reviewers. I am suggesting a couple of very small additions that relate to some of the suggestions made by the reviewers which I think will benefit the manuscript:

Relating to the comment by reviewer 1 on the OH experimental budget, I think it would be useful to add the product of the measured OH concentration and measured OH reactivity (kOH) to Fig. S3(a) and also show Poh – (kOH*[OH]) in Fig. S3(b).

As suggested, we have added the total OH loss rate calculated from measured concentrations of OH and the measured OH reactivity to Figure S3a and also the net rate of production or loss, based on the measured reactivity, in Figure S3b.

[Figure]

For comment 1 by reviewer 2, I suggest including the monoterpene concentration in Table S3 alongside the isoprene concentration (if measurements were made). This could then be referred to during the discussion on modelled and measured nighttime XO2 concentrations. It is not clear from the discussion currently if the models for the earlier campaigns were constrained to monoterpenes or not?

We have added monoterpene measurements to Table S3. The measured mixing ratios were similar in 1998, 2008, 2009, and 2016, but unfortunately monoterpenes were not measured in 1997 during the only other campaign in which total peroxy radicals were measured.

Although not measured at the site during the 1997 campaign, monoterpene mixing ratios observed in 1998, 2008, and 2009 were similar to measurements from 2016 (Table S3).

We have also added a reference to an additional study which compares the total peroxy radical measurements from 1997 to a model that includes reactive terpene emissions. The measured and modeled total peroxy radical concentrations in that study were very similar to our results from 2016. We've added this comparison that focuses on the discrepancy between the measurements and models at night to page 18 of the revised manuscript.

The RACM2 and MCM models overpredict the nighttime $XO_2$ concentrations by factor of approximately 4, with the RACM2-LIM1 model predicting mixing ratios of total peroxy radicals of approximately 27 ppt between 21:00 and 6:00 and the MCM v3.3.1 model predicting $XO_2$ mixing ratios of approximately 36 ppt during the night (Fig. 4) compared to the measured concentrations of less than 10 ppt. These results are similar to those from the 1997 PROPHET campaign in which measured $XO_2$ mixing ratios of 3–6 ppt were overpredicted by more than a factor of 10 by a model that included reactive terpene emissions (Mihele and Hastie, 2003; Sillman et al., 2002).

**References**

Sillman, S., Carroll, M. A., Thornberry, T., Lamb, B. K., Westberg, H., Brune, W. H., Faloona, I., Tan, D., Shepson, P. B., Sumner, A. L., Hastie, D. R., Mihele, C. M., Apel, E. C., Riemer, D. D., and Zika, R. G.: Loss of isoprene and sources of nighttime OH radicals at a rural site in the United States: Results from photochemical models, J. Geophys. Res.- Atmos., 107, ACH 2-1-ACH 2-14, https://doi.org/https://doi.org/10.1029/2001JD000449, 2002.